# Generic Neural Architecture Search via Regression

**Yuhong Li**[1], **Cong Hao**[2], **Pan Li**[3], **Jinjun Xiong**[4], **Deming Chen**[1]
University of Illinois at Urbana-Champaign[1], Georgia Institute of Technology[2],
Purdue University[3], University at Buffalo [4]
leeyh@illinois.edu, callie.hao@ece.gatech.edu, panli@purdue.edu,
jinjun@buffalo.edu, dchen@illinois.edu

## Abstract

Most existing neural architecture search (NAS) algorithms are dedicated to and evaluated by the downstream tasks, e.g., image classification in computer vision. However, extensive experiments have shown that, prominent neural architectures, such as ResNet in computer vision and LSTM in natural language processing, are generally good at extracting patterns from the input data and perform well on different downstream tasks. In this paper, we attempt to answer two fundamental questions related to NAS. (1) Is it necessary to use the performance of specific downstream tasks to evaluate and search for good neural architectures? (2) Can we perform NAS effectively and efficiently while being agnostic to the downstream tasks? To answer these questions, we propose a novel and generic NAS framework, termed **Gen**eric **NAS** (GenNAS). GenNAS does not use task-specific labels but instead adopts *regression* on a set of manually designed synthetic signal bases for architecture evaluation. Such a self-supervised regression task can effectively evaluate the intrinsic power of an architecture to capture and transform the input signal patterns, and allow more sufficient usage of training samples. Extensive experiments across 13 CNN search spaces and one NLP space demonstrate the remarkable efficiency of GenNAS using regression, in terms of both evaluating the neural architectures (quantified by the ranking correlation Spearman's $\rho$ between the approximated performances and the downstream task performances) and the convergence speed for training (within a few seconds). For example, on NAS-Bench-101, GenNAS achieves 0.85 $\rho$ while the existing efficient methods only achieve 0.38. We then propose an automatic task search to optimize the combination of synthetic signals using limited downstream-task-specific labels, further improving the performance of GenNAS. We also thoroughly evaluate Gen-NAS's generality and end-to-end NAS performance on all search spaces, which outperforms almost all existing works with significant speedup. For example, on NASBench-201, GenNAS can find near-optimal architectures within 0.3 GPU hour. Our code has been made available at: *https://github.com/leeyeehoo/GenNAS*

## 1 Introduction

Most existing neural architecture search (NAS) approaches aim to find top-performing architectures on a specific downstream task, such as image classification [1, 2, 3, 4, 5], semantic segmentation [6, 7, 8], neural machine translation [9, 10, 11] or more complex tasks like hardware-software co-design [12, 13, 14, 15, 16]. They either directly search on the target task using the target dataset (e.g., classification on CIFAR-10 [2, 17] ), or search on a *proxy* dataset and then transfer to the target one (e.g. CIFAR-10 to ImageNet) [18, 3]. However, extensive experiments show that prominent neural architectures are generally good at extracting patterns from the input data and perform well to different downstream tasks. For example, ResNet [19] being a prevailing architecture in computer vision, shows outstanding performance across various datasets and tasks [20, 21, 22], because of its

35th Conference on Neural Information Processing Systems (NeurIPS 2021).

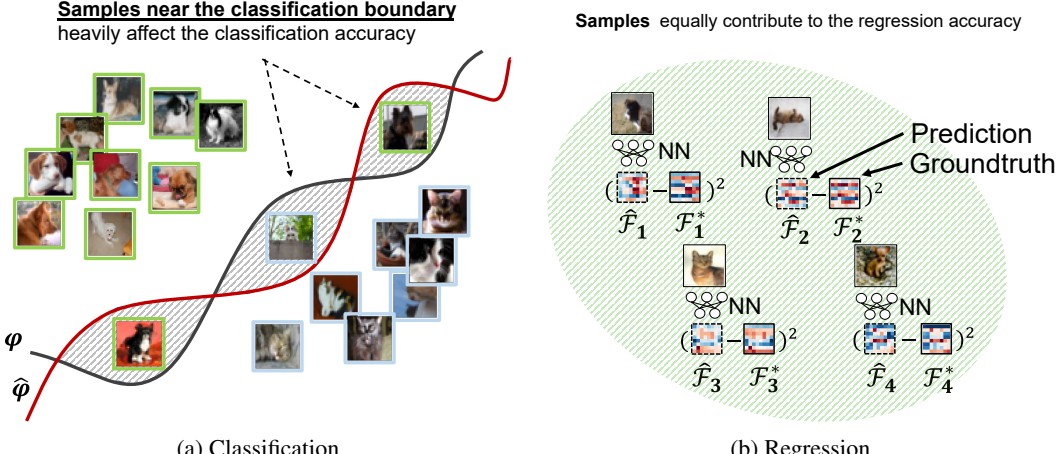

(a) Classification                 (b) Regression

Figure 1: For classification, only samples near the decision boundary determine the classification accuracy. For regression, all samples equally contribute to the regression accuracy. Therefore, regression is better at leveraging all training samples than classification to achieve faster convergence.

advantageous architecture, the residual blocks. This observation motivates us to ask the first question: *Is there a generic way to search for and evaluate neural architectures without using the specific knowledge of downstream tasks?*

Meanwhile, we observe that most existing NAS approaches directly use the *final classification performance* as the metric for architecture evaluation and search, which has several major issues. First, the classification accuracy is dominated by the samples along the classification boundary, while other samples have clearer classification outcomes compared to the boundary ones (as illustrated in Fig. 1a). Such phenomena can be observed in the limited number of effective support vectors in SVM [23], which also applies to neural networks because of the theory of neural tangent kernel [24]. Therefore, discriminating performance of classifiers needs many more samples than necessary (the indeed effective ones), causing a big waste. Second, a classifier tends to discard a lot of valuable information, such as finer-grained features and spatial information, by transforming input representations into categorical labels. This observation motivates us to ask the second question: *Is there a more effective way that can make more sufficient use of input samples and better capture valuable information?*

To answer the two fundamental questions for NAS, in this work, we propose a **Gen**eric **N**eural **A**rchitecture **S**earch method, termed **GenNAS**. GenNAS adopts a **regression-based proxy task** using **downstream-task-agnostic synthetic signals** for network training and evaluation. It can efficiently (with near-zero training cost) and accurately *approximate* the neural architecture performance.

**Insights**. First, as opposed to classification, regression can efficiently make fully use of all the input samples, which equally contribute to the regression accuracy (Fig. 1b). Second, regression on properly-designed synthetic signals is essentially evaluating the *intrinsic representation power* of neural architectures, which is to capture and distinguish fundamental data patterns that are agnostic to downstream tasks. Third, such representation power is heavily reflected in the *intermediate data* of a network (as we will show in the experiments), which are regrettably discarded by classification.

**Approach**. First, we propose a *regression proxy task* as the supervising task to train, evaluate, and search for neural architectures (Fig. 2). Then, the searched architectures will be used for the target downstream tasks. To the best of our knowledge, we are the first to propose self-supervised regression proxy task instead of classification for NAS. Second, we propose to use *unlabeled synthetic data* (e.g., sine and random signals) as the groundtruth (Fig. 3) to measure neural architectures' intrinsic capability of capturing fundamental data patterns. Third, to further boost NAS performance, we propose a weakly-supervised automatic proxy task search with only a handful of groundtruth architecture performance (e.g. 20 architectures), to determine the best proxy task, i.e., the combination of synthetic signal bases, targeting a specific downstream task, search space, and/or dataset (Fig. 4).

**GenNAS Evaluation**. The efficiency and effectiveness of NAS are dominated by *neural architecture evaluation*, which directs the search algorithm towards top-performing network architectures. To quantify how accurate the evaluation is, one widely used indicator is the network performance *Ranking*

*Correlation* [25] between the prediction and groundtruth ranking, defined as Spearman's Rho ($\rho$) or Kendall's Tau ($\tau$). The ideal ranking correlation is 1 when the approximated and groundtruth rankings are exactly the same; achieving large $\rho$ or $\tau$ can significantly improve NAS quality [26, 27, 28]. Therefore, in the experiments (Sec. 4), we evaluate GenNAS using the ranking correlation factors it achieves, and then show its end-to-end NAS performance in finding the best architectures. Extensive experiments are done on 13 CNN search spaces and one NLP space [29]. Trained by the regression proxy task using only a single batch of unlabeled data within a few seconds, GenNAS significantly outperforms all existing NAS approaches on almost all the search spaces and datasets. For example, GenNAS achieves 0.87 $\rho$ on NASBench-101 [30], while Zero-Cost NAS [31], an efficient proxy NAS approach, only achieves 0.38. On end-to-end NAS, GenNAS generally outperforms others with large speedup. This implies that the insights behind GenNAS are plausible and that our proposed regression-based task-agnostic approach is generalizable across tasks, search spaces, and datasets.

**Contributions**. We summarize our contributions as follows:

- To the best of our knowledge, GenNAS is the first NAS approach using regression as the self-supervised proxy task instead of classification for neural architecture evaluation and search. It is agnostic to the specific downstream tasks and can significantly improve training and evaluation efficiency by fully utilizing only a handful of unlabeled data.

- GenNAS uses synthetic signal bases as the groundtruth to measure the intrinsic capability of networks that captures fundamental signal patterns. Using such unlabeled synthetic data in regression, GenNAS can find the generic task-agnostic top-performing networks and can apply to any new search spaces with zero effort.

- An automated proxy task search to further improve GenNAS performance.

- Thorough experiments show that GenNAS outperforms existing NAS approaches by large margins in terms of ranking correlation with near-zero training cost, across 13 CNN and one NLP space *without* proxy task search. GenNAS also achieves state-of-the-art performance for end-to-end NAS with orders of magnitude of speedup over conventional methods.

- With proxy task search being optional, GenNAS is fine-tuning-free, highly efficient, and can be easily implemented on a single customer-level GPU.

## 2    Related Work

**NAS Evaluation.** Network architecture evaluation is critical in guiding the search algorithms of NAS by identifying the top-performing architectures, which is also a challenging task with intensive research interests. Early NAS works evaluated the networks by training from scratch with tremendous computation and time cost [18, 1]. To expedite, weight-sharing among the subnets sampled from a supernet is widely adopted [3, 28, 4, 32, 33]. However, due to the poor correlation between the weight-sharing and the final performance ranking, weight-sharing NAS can easily fail even in simple search spaces [34, 35]. Yu et al. [36] further pointed out that without accurate evaluation, NAS runs in a near-random fashion. Recently, zero-cost NAS methods [37, 31, 38, 39] have been proposed, which score the networks using their initial parameters with only one forward and backward propagation. Despite the significant speed up, they fail to identify top-performing architectures in large search spaces such as NASBench-101. To detach the time-consuming network evaluation from NAS, several benchmarks are developed with fully-trained neural networks within the NAS search spaces [40, 30, 35, 29, 41], so that researchers can assess the search algorithms alone in the playground.

**NAS Transferability.** To improve search efficiency, proxy tasks are widely used, on which the architectures are searched and then transferred to target datasets and tasks. For example, the CIFAR-10 classification dataset seems to be a good proxy for ImageNet [18, 3]. Kornblith et al. [42] studied the transferability of 16 classification networks on 12 image classification datasets. NASBench-201 [35] evaluated the ranking correlations across three popular datasets with 15625 architectures. Liu et al. [43] studied the architecture transferability across supervised and unsupervised tasks. Nevertheless, training on a downsized proxy dataset is still inefficient (e.g. a few epochs of full-blown training [43]). In contrast, GenNAS significantly improves the efficiency by using a single batch of data while maintaining extremely good generalizability across different search spaces and datasets.

**Self-supervised Learning.** Self-supervised learning is a form of unsupervised learning, that the neural architectures are trained with automatically generated labels to gain a good degree of com-

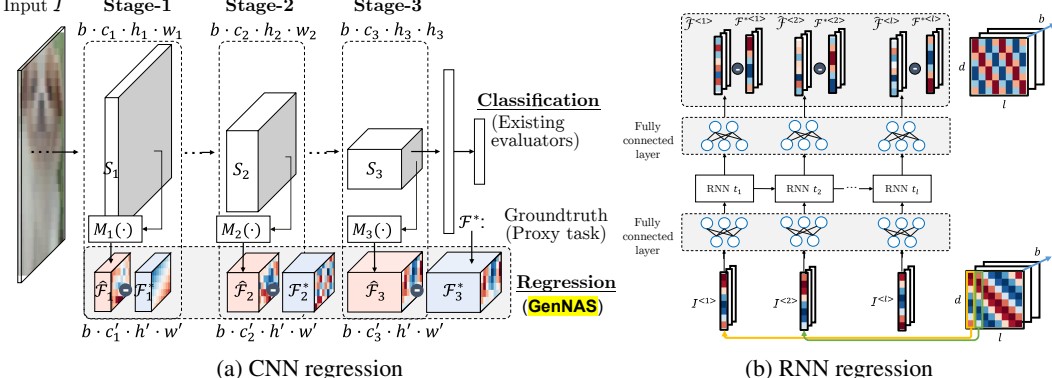

| (a) CNN regression | (b) RNN regression |

Figure 2: Regression architectures on CNNs and RNNs. (a) On CNNs, we remove the final classifier and extract multiple stages of intermediate feature map for training. (b) On RNNs, we construct a many-to-many regression task, where the input and output tensors have the same size.

prehension or understanding [44, 45, 46, 47, 43]. Liu et al. [43] recently proposed three unlabeled classification proxy tasks, including rotation prediction, colorization, and solving jigsaw puzzles, for neural network evaluation. Though promising, this approach did not explain why such manually designed proxy tasks are beneficial and still used classification for training with the entire dataset. In contrast, GenNAS uses regression with only a single batch of synthetic data.

## 3  Proposed GenNAS

In Section 3.1, we introduce the main concepts of task-agnostic GenNAS: 1) the proposed regression proxy task for both CNN architectures and recurrent neural network (RNN) architectures; 2) the synthetic signal bases used for representing the fundamental data patterns as the proxy task. In Section 3.2, we introduce the automated proxy task search.

### 3.1  GenNAS

#### 3.1.1  Regression Architectures

Training using unlabeled regression is the key that GenNAS being agnostic to downstream tasks. Based on the insights discussed in Section 1, the *principle* of designing the regression architecture is to *fully utilize the abundant intermediate information* rather than the final classifier.

**Regression on CNNs.** Empirical studies show that CNNs learn fine-grained high-frequency spatial details in the early layers and produce semantic features in the late layers [48]. Following this principle, as shown in Fig. 2a, we construct a Fully Convolutional Network (FCN) [49] by removing the final classifier of a CNN, and then extract the FCN's intermediate feature maps from *multiple stages*. We denote the number of stages as $N$. **Inputs**. The inputs to the FCN are unlabeled real images, shaped as a tensor $\mathcal{I} \in \mathbb{R}^{b \times 3 \times h \times w}$, where $b$ is the batch size, and $h$ and $w$ are the input image size. **Outputs**. From each stage $i$ ($1 \leq i \leq N$) of the FCN, we first extract a feature map tensor, denoted by $\mathcal{F}_i \in \mathbb{R}^{b \times c_i \times h_i \times w_i}$, and reshape it as $\hat{\mathcal{F}}_i \in \mathbb{R}^{b \times c_i' \times h' \times w'}$ through a convolutional layer $M_i$ by $\hat{\mathcal{F}}_i = M_i(\mathcal{F}_i)$ (with downsampling if $w_i > w'$ or $h_i > h'$). The outputs are the tensors $\hat{\mathcal{F}}_i$, which encapsulate the captured signal patterns from different stages. **Groundtruth**. We construct a synthetic signal tensor for each stage as the groundtruth, which serves as part of the *proxy task*. A synthetic tensor is a combination of multiple synthetic signal bases (more details in Section 3.1.2), denoted by $\mathcal{F}_i^*$. We compare $\hat{\mathcal{F}}_i$ with $\mathcal{F}_i^*$ for training and evaluating the neural architectures. During training, we use MSE loss defined as $\mathcal{L} = \sum_{i=1}^{N} \mathbf{E}[(\mathcal{F}_i^* - \hat{\mathcal{F}}_i)^2]$; during validation, we adjust each stage's output importance as $\mathcal{L} = \sum_{i=1}^{N} \frac{1}{2^{N-i}} \mathbf{E}[(\mathcal{F}_i^* - \hat{\mathcal{F}}_i)^2]$ since the feature map tensors of later stages are more related to the downstream task's performance. The detailed configurations of $N$, $h'$, $w'$, and $c_i'$ are provided in the experiments.

**Regression on RNNs.** The proposed regression proxy task can be similarly applied to NLP tasks using RNNs. Most existing NLP models use a sequence of word-classifiers as the final outputs,

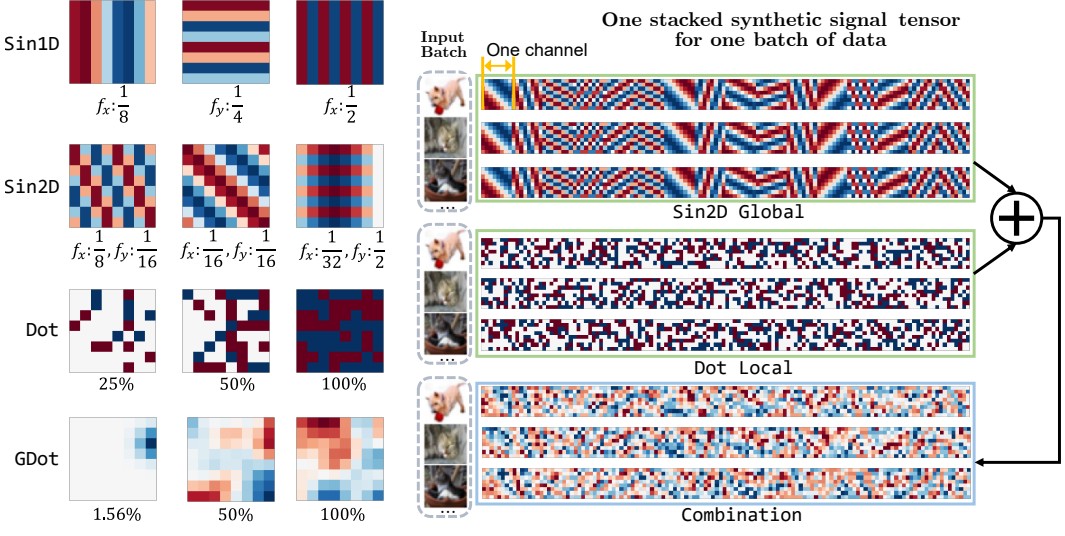

(a) Examples of signal bases       (b) Examples of synthetic signal tensors

Figure 3: (a) Examples of synthetic signal bases (2D feature maps). (b) Examples of the synthetic signal tensors by stacking 2D feature maps along the channel dimension for CNN architectures.

whose evaluations are thus based on the word classification accuracy [50, 51, 52]. Following the same principle for CNNs, we design a many-to-many regression task for RNNs as shown in Fig. 2b. Instead of using the final word-classifier's output, we extract the output tensor of the intermediate layer before it. **Inputs**. For a general RNN model, the input is a random tensor $\mathcal{I} \in \mathbb{R}^{l \times b \times d}$, where $l$ is the sequence length, $b$ is the batch size, and $d$ is the length of input/output word vectors. Given a sequence of length $l$, the input to the RNN each time is one slice of the tensor $\mathcal{I}$, denoted by $\mathcal{I}^{(i)} \in \mathbb{R}^{b \times d}$, $1 \leq i \leq l$. **Outputs**. The output is $\hat{\mathcal{F}} \in \mathbb{R}^{l \times b \times d}$, where a slice of $\hat{\mathcal{F}}$ is $\hat{\mathcal{F}}^{(i)} \in \mathbb{R}^{b \times d}$. **Groundtruth**. Similar to the CNN case, we generate a synthetic signal tensor $\mathcal{F}^*$ as the proxy task groundtruth.

### 3.1.2 Synthetic Signal Bases

The proxy task for regression aims to capture the task-agnostic intrinsic learning capability of the neural architectures, i.e., representing various fundamental data patterns. For example, good CNNs must be able to learn different frequency signals to capture image features [53]. Here, we design four types of synthetic signal basis: (1) 1-D frequency basis (Sin1D); (2) 2-D frequency basis (Sin2D); (3) Spatial basis (Dot and GDot); (4) Resized input signal (Resize). Sin1D and Sin2D represent frequency information, Dot and GDot represent spatial information, and Resize reflects the CNN's scale-invariant capability. The combinations of these signal bases, especially with different sine frequencies, can represent a wide range of complicated real-world signals [54]. If a network architecture is good at learning such signal basis and their simple combinations, it is more likely to be able to capture real-world signals from different downstream tasks.

Fig.3a depicts examples of synthetic signal bases, where each base is a 2D signal feature map. Sin1D is generated by $\sin(2\pi f x + \phi)$ or $\sin(2\pi f y + \phi)$, and Sin2D is generated by $\sin(2\pi f_x x + 2\pi f_y y + \phi)$, where $x$ and $y$ are pixel indices. Dot is generated according to biased Rademacher distribution [55] by randomly setting $k\%$ pixels to $\pm 1$ on zeroed feature maps. GDot is generated by applying a Gaussian filter with $\sigma = 1$ on Dot and normalizing between $\pm 1$. The synthetic signal tensor $\mathcal{F}^*$ (the proxy task groundtruth) is constructed by stacking the 2D signal feature maps along the channel dimension (CNNs) or the batch dimension (for RNNs). Fig.3b shows examples of stacked synthetic tensor $\mathcal{F}^*$ for CNN architectures. Within one batch of input images, we consider two settings: global and local. The global setting means that the synthetic tensor is the same for all the inputs within the batch, as the Sin2D Global in Fig. 3b, aiming to test the network's ability to capture invariant features from different inputs; the local setting uses different synthetic signal tensors for different inputs, as the Dot Local in Fig.3b, aiming to test the network's ability to distinguish between images. For CNNs, the real images are only used by resize, and both global and local settings are used. For RNNs, we only use synthetic signals and the local setting, because resizing natural language or time series, the typical input of RNNs, does not make as much sense as resizing images for CNNs.

## 3.2 Proxy Task Search

While the synthetic signals can express generic features, the importance of these features for different tasks, NAS search spaces, and datasets may be different. Therefore, we further propose a weakly-supervised proxy task search, to automatically find the best synthetic signal tensor, i.e., the best combination of synthetic signal bases. We define the *proxy task search space* as the parameters when generating the synthetic signal tensors. As illustrated in Fig. 4, first, we randomly sample a small subset (e.g., 20) of the neural architectures in the NAS search space and obtain their groundtruth ranking on the target task (e.g., image classification). We then train these networks using different proxy tasks and calculate the performance ranking correlation $\rho$ of the proxy and the target task. We use the regularized tournament selection evolutionary algorithm [1] to search for the task that results in the largest $\rho$, where $\rho$ is the fitness function.

**Proxy Task Search Space.** We consider the following parameters as the proxy task search space. (1) Noise. We add noise to the input data following the distribution of parameterized Gaussian or uniform distribution. (2) The number of channels for each synthetic signal tensor ($c_i$ in $\mathcal{F}_i^* \in \mathbb{R}^{b \times c_i \times h' \times w'}$) can be adjusted. (3) Signal parameters, such as the frequency $f$ and phase $\phi$ in Sin, can be adjusted. (4) Feature combination. Each synthetic signal tensor uses either local or global, and tensors can be selected and summed up. Detailed parameters can be found in the supplemental material.

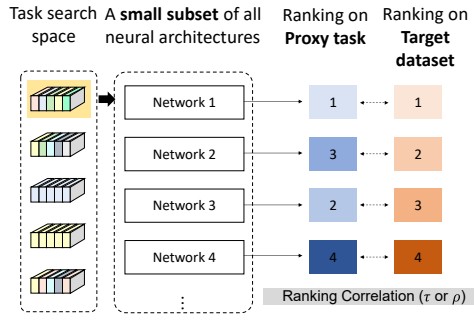

Figure 4: Proxy task search.

## 4 Experiments

We perform the following evaluations for GenNAS. First, to show the *true power of regression*, we use manually designed proxy tasks *without task search* and apply the same proxy task on all datasets and search spaces. We demonstrate that the *GenNAS generally excels in all different cases with zero task-specific cost*, thanks to unlabeled self-supervised regression proxy task. Specifically, in Section 4.1, we analyze the effectiveness of the synthetic signal bases and manually construct two sets of synthetic tensors as the baseline proxy tasks; in Section 4.2, we extensively evaluate the proposed regression approach in 13 CNN search spaces and one NLP search space. Second, in Section 4.3, we evaluate the proxy task search and demonstrate the remarkable generalizability by applying one searched task to all NAS search spaces with no change. Third, in Section 4.4, we evaluate GenNAS on end-to-end NAS tasks, which outperforms existing works with significant speedup.

**Experiment Setup**. We consider 13 CNN NAS search spaces including NASBench-101 [30], NASBench-201 [35], Network Design Spaces (NDS) [56], and one NLP search space, NASBench-NLP [29]. All the training is conducted using only one batch of data with batch size 16 for 100 iterations. Details of NAS search spaces and experiment settings are in the supplemental material.

### 4.1 Effectiveness of Synthetic Signals

The synthetic signal analysis is performed on NASBench-101 using CIFAR-10 dataset. From the whole NAS search space, 500 network architectures are randomly sampled with a known performance ranking provided by NASBench-101. We train the 500 networks using different synthetic signal tensors and calculate their ranking correlations with respect to the groundtruth ranking. Using the CNN architecture discussed in Section 3.1.1, we consider three stages, $S_1$ to $S_3$ for $N = 3$; the number of channels is 64 for each stage. For Sin1D and Sin2D, we set three ranges for frequency $f$: low (L) $f \in (0, 0.125)$, medium (M) $f \in (0.125, 0.375)$, and high (H) $f \in (0.375, 0.5)$. Within each frequency range, 10 signals are generated using uniformly sampled frequencies. For Dot and GDot, we randomly set $50\%$ and $100\%$ pixels to $\pm 1$ on the zeroized feature maps.

The results of ranking correlations are shown in Table 1. The three stages are evaluated independently and then used together. Among the three stages, Sin1D and Sin2D within medium and high frequency work better in $S_1$ and $S_2$, while the high frequency Dot and resize work better in $S_3$. The low frequency signals, such as GDot, Sin1D-L, Sin2D-L, and the extreme case zero tensors, result in low ranking correlations; we attribute to their poor distinguishing ability. We also observe that the

Table 1: Ranking correlation (Spearman's $\rho$) analysis of different synthetic signals on NASBench-101.

| Stage | Sin1D | | | Sin2D | | | Dot | | GDot | | Resize | Zero |
|---|---|---|---|---|---|---|---|---|---|---|---|---|
| | L | M | H | L | M | H | 50% | 100% | 50% | 100% | | |
| $S_1$ | 0.13 | 0.43 | **0.64** | 0.14 | 0.53 | 0.63 | 0.55 | 0.62 | 0.18 | 0.16 | 0.56 | 0.17 |
| $S_2$ | 0.03 | 0.52 | **0.79** | 0.05 | 0.73 | 0.72 | 0.64 | 0.69 | 0.03 | 0.02 | 0.73 | 0.18 |
| $S_3$ | 0.08 | 0.77 | 0.80 | 0.23 | 0.78 | 0.72 | 0.76 | **0.81** | 0.16 | 0.17 | 0.80 | 0.22 |
| **GenNAS-combo:: 0.85** | | | | | | | | | | | | |

best task in $S_3$ (0.81) achieves higher $\rho$ than $S_1$ (0.64) and $S_2$ (0.79), which is consistent with the intuition that the features learned in deeper stages have more impact to the final network performance.

When all three stages are used, where each stage uses its top-3 signal bases, the ranking correlation can achieve 0.85, higher than the individual stages. This supports our assumption in Section 3.1.1 that utilizing more intermediate information of a network is beneficial. From this analysis, we choose two top-performing proxy tasks in the following evaluations to demonstrate the effectiveness of regression: **GenNAS-single** – the best proxy task with a single signal tensor `Dot%100` used only in $S_3$, and **GenNAS-combo** – the combination of the three top-performing tasks in three stages.

### 4.2 Effectiveness and Efficiency of Regression without Proxy Task Search

To quantify how well the proposed regression can approximate the neural architecture performance with only one batch of data within seconds, we use the ranking correlation, Spearman's $\rho$, as the metric [31, 43, 34]. We use the two manually designed proxy tasks (GenNAS-single and GenNAS-combo) without proxy task search to demonstrate that **GenNAS is generic and can be directly applied to any new search spaces with zero task-specific search efforts**. The evaluation is extensively conducted on 13 CNN search spaces and 1 NLP search space, and the results are summarized in Table 2.

On NASBench-101, GenNAS is compared with zero-cost NAS [31, 37] and the latest classification based approaches [43]. Specifically, NASWOT [37] is a zero-training approach that predicts a network's trained accuracy from its initial state by examining the overlap of activations between datapoints. Abdelfattah et al. [31] proposed proxies such as synflow to evaluate the networks, where the synflow computes the summation of all the weights multiplied by their gradients and has the best reported performance in the paper. Liu et al. [43] used three unsupervised classification training proxies, namely rotation prediction (rot), colorization (col), and solving jigsaw puzzles (jig), and one supervised classification proxy (cls).

We report their results after 10 epochs (@ep10) for each proxy. The results show that GenNAS-single and GenNAS-combo achieve 0.81 and 0.85 $\rho$ on CIFAR-10, and achive 0.73 on ImageNet, respectively, much higher than NAS-WOT and synflow. It is also comparable and even higher comparing with the classification proxies, cls@ep5 and cls@ep10. Notably, the classification proxies need to train for 10 epochs using *all training data*, while GenNAS requires only a few seconds, more than $40\times$ faster. On NASBench-201, we further compare with vote [31] and EcoNAS [26]. EcoNAS is a recently proposed reduced-training proxy NAS approach. Vote [31] adopts the majority vote between three zero-training proxies including synflow, jacob_cov, and snip. Clearly, GenNAS-combo outperforms all these methods regarding ranking correlation, and is also $60\times$ faster than EcoNAS and $40\times$ faster than cls@ep10.

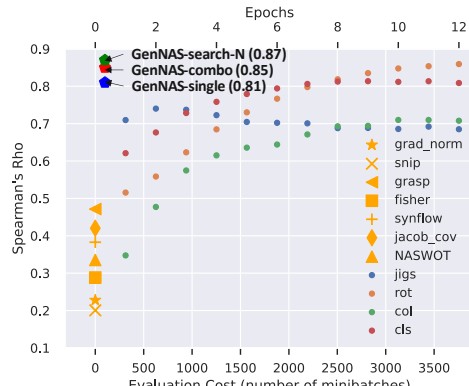

Figure 5: The effectiveness of regression-based proxy task. GenNAS significantly outperforms all the existing NAS evaluation approaches regarding ranking correlation, with near-zero training cost.

On Neural Design Spaces, we evaluate GenNAS on both CIFAR-10 and ImageNet datasets. Comparing with NASWOT and synflow, GenNAS-single and GenNAS-combo achieve higher $\rho$ in almost all cases. Also, synflow performs poorly on most of the NDS search spaces especially on ImageNet dataset, while GenNAS achieves even higher $\rho$. Extending to NLP search space, NASBench-NLP, GenNAS-single and GenNAS-combo achieve 0.73 and 0.74 $\rho$, respectively, surpassing the best

Table 2: GenNAS ranking correlation evaluation using the correlation Spearman's $\rho$. **GenNAS-single** and **GenNAS-combo** use a single or a combination of synthetic signals that are manually designed *without proxy task search*. **GenNAS search-N, -D, -R** mean the proxy task is searched on NASBench-101, NDS DARTS design space, and NDS ResNet design space, respectively. The top-1/2/3 results of GenNAS and efficient NAS baselines are highlighted by †/‡/§ respectively for each task. The values with superscripts are obtained after task search (s) or transferred (t) from a previous searched task. Methods like jig@ep10 which is 40x slower compared to the GenNAS in prediction are not considered as efficient ones.

**NASBench-101**

| Dataset | NASWOT [37] | synflow [31] | jig@ep10 | rot@ep10 | col@ep10 | cls@ep10 | GenNAS single | combo | search-N |
|---|---|---|---|---|---|---|---|---|---|
| | | | > 40× slower | | | | | | |
| CIFAR-10 | 0.34 | 0.38 | 0.69 | 0.85 | 0.71 | 0.81 | 0.81§ | 0.85‡ | **0.87**†s |
| ImgNet | 0.21 | 0.09 | 0.72 | 0.82 | 0.67 | 0.79 | 0.65§ | **0.73**† | 0.71‡t |

**NASBench-201**

| Dataset | NASWOT | synflow | jacob_cov | snip | cls@ep10 | vote | EcoNAS | GenNAS single | combo | search-N |
|---|---|---|---|---|---|---|---|---|---|---|
| | | | | | > 40× slower | | > 60× slower | | | |
| CIFAR-10 | 0.79§ | 0.72 | 0.76 | 0.57 | 0.75 | 0.81 | 0.81 | 0.77 | 0.87‡ | **0.90**†t |
| CIFAR-100 | 0.81 | 0.76 | 0.70 | 0.61 | 0.75 | 0.83‡ | 0.75 | 0.69 | 0.82§ | **0.84**†t |
| ImgNet16 | 0.78 | 0.73 | 0.73 | 0.59 | 0.68 | 0.81§ | 0.77 | 0.70 | 0.81‡ | **0.87**†t |

**Neural Design Spaces**

| Dataset | NAS-Space | NASWOT | synflow | cls@ep10 | GenNAS single | combo | search-N | search-D | search-R |
|---|---|---|---|---|---|---|---|---|---|
| | | | | > 40× slower | | | | | |
| CIFAR-10 | DARTS | 0.65 | 0.41 | 0.63 | 0.43 | 0.68 | 0.71§t | **0.86**†s | 0.82‡t |
| | DARTS-f | 0.31 | 0.09 | 0.82 | 0.51 | **0.59**† | 0.53‡t | 0.58‡t | 0.52t |
| | Amoeba | 0.33 | 0.06 | 0.67 | 0.52 | 0.64 | 0.68§t | **0.78**†t | 0.72‡t |
| | ENAS | 0.55 | 0.19 | 0.66 | 0.56 | 0.70§ | 0.67t | **0.82**†t | 0.78‡t |
| | ENAS-f | 0.43 | 0.26 | 0.86 | 0.65 | 0.65 | 0.67§t | **0.73**†t | 0.67‡t |
| | NASNet | 0.40 | 0.00 | 0.64 | 0.56 | 0.66§ | 0.65t | **0.77**†t | 0.71‡t |
| | PNAS | 0.51 | 0.26 | 0.50 | 0.32 | 0.58 | 0.59§t | **0.76**†t | 0.71‡t |
| | PNAS-f | 0.10 | 0.32 | 0.85 | 0.45 | 0.48§ | **0.56**†t | 0.55‡t | 0.47t |
| | ResNet | 0.26 | 0.22 | 0.65 | 0.34 | 0.52 | 0.55§t | 0.54‡t | **0.83**†s |
| | ResNeXt-A | 0.65§ | 0.48 | 0.86 | 0.57 | 0.61 | 0.80‡t | 0.63t | **0.84**†t |
| | ResNeXt-B | 0.60§ | 0.60‡ | 0.66 | 0.26 | 0.30 | 0.53t | 0.55t | **0.71**†t |
| ImageNet | DARTS | 0.66 | 0.21 | – | 0.61 | 0.75‡ | 0.70§t | **0.84**†t | 0.55t |
| | DARTS-f | 0.20 | 0.37 | – | 0.68§ | 0.69‡ | 0.67t | **0.69**†t | 0.59t |
| | Amoeba | 0.42 | 0.25 | – | 0.63 | 0.72§ | 0.73‡t | **0.80**†t | 0.67t |
| | ENAS | 0.69§ | 0.17 | – | 0.59 | 0.70‡ | 0.58t | **0.81**†t | 0.65t |
| | NASNet | 0.51 | 0.01 | – | 0.52 | 0.59§ | 0.52t | **0.70**†t | 0.61‡t |
| | PNAS | 0.60‡ | 0.14 | – | 0.28 | 0.39 | 0.45§t | **0.62**†t | 0.40t |
| | ResNeXt-A | 0.72 | 0.42 | – | 0.80§ | 0.84‡ | 0.75t | 0.62t | **0.87**†t |
| | ResNeXt-B | 0.63 | 0.31 | – | 0.71‡ | **0.79**† | 0.51t | 0.60t | 0.64§t |

**NASBench-NLP**

| Dataset | grad_norm | snip | grasp | synflow | jacob_cov | ppl@ep3 | GenNAS single | combo | search |
|---|---|---|---|---|---|---|---|---|---|
| | | | | | | > 192× slower | | | |
| PTB | 0.21 | 0.19 | 0.16 | 0.34 | 0.56§ | 0.79 | 0.73 | 0.74‡ | **0.81**†s |

zero-proxy method (0.56). Comparing with the ppl@ep3, the architectures trained on PTB [57] dataset after three epochs, GenNAS is 192× faster in prediction.

Fig. 5 visualizes the comparisons between GenNAS and existing NAS approaches on NASBench-101, CIFAR-10. Clearly, regression-based GenNAS (single, combo) significantly outperforms the existing NAS with near-zero training cost, showing remarkable effectiveness and high efficiency.

## 4.3 Effectiveness of Proxy Task Search and Transferability

**Effectiveness of Proxy Task Search.** While the *unsearched* proxy tasks can already significantly outperform all existing approaches (shown in Section 4.2), we demonstrate that the proxy task search described in Section 3.2 can further improve the ranking correlation. We adopt the regularized evolutionary algorithm [1]. The population size is 50; the tournament sample size is 10; the search runs 400 iterations. We randomly select 20 architectures with groundtruth for calculating the $\rho$. More settings can be found in the supplemental material. Fig. 6 shows the search results averaged from 10 runs with different seeds. It shows that the regularized evolutionary algorithm is more effective comparing with random search, where the correlations of 20 architectures are $0.86 \pm 0.02$ and $0.82 \pm 0.01$, respectively.

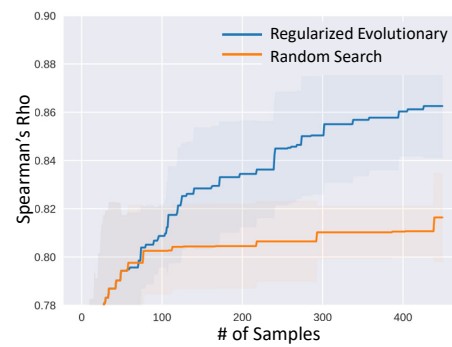

Figure 6: Proxy task search.

Table 3: GenNAS end-to-end NAS results comparing with the state-of-the-art NAS approaches, showing test accuracy (%) on different NAS-spaces and datasets. $^\star$ denotes a method that is replicated with the same regularized evolutionary algorithm in Section 4.4 for fair comparison. On NASBench-201, the GPU hours do not include task search since GenNAS-N is transferred from NASBench-101. The values with superscripts are obtained after task search ($^s$) or transferred ($^t$) from a previous searched task.

**NASBench-101(CIFAR-10)**

| Optimal | NASWOT$^\star$ | synflow$^\star$ | Halfway$^\star$ | GenNAS-N |
|---|---|---|---|---|
| 94.32 | 93.30±0.002 | 91.31±0.02 | 93.28±0.002 | **93.92±0.004**$^s$ |

**NASBench-201**

| Dataset | Optimal | RSPS | DARTS-V2 | GDAS | SETN | ENAS | NASWOT | GenNAS-N |
|---|---|---|---|---|---|---|---|---|
| CIFAR-10 | 94.37 | 84.07±3.61 | 54.30±0.00 | 93.61±0.09 | 87.64±0.00 | 53.89±0.58 | 92.96±0.80 | **94.18±0.10**$^t$ |
| CIFAR-100 | 73.49 | 58.33±4.34 | 15.61±0.00 | 70.61±0.26 | 56.87±7.77 | 15.61±0.00 | 70.03±1.16 | **72.56±0.74**$^t$ |
| ImgNet16 | 47.31 | 26.28±3.09 | 16.32±0.00 | 41.71±0.98 | 32.52±0.21 | 14.84±2.10 | 44.43±2.07 | **45.59±0.54**$^t$ |
| GPU hours | | 2.2 | 9.9 | 8.8 | 9.5 | 3.9 | 0.1 | 0.3 |

**Neural Design Spaces (CIFAR-10)**

| NAS-Space | Optimal | NASWOT$^\star$ | synflow$^\star$ | cls@ep10$^\star$ | GenNAS-N | GenNAS-R | GenNAS-D |
|---|---|---|---|---|---|---|---|
| ResNet | 95.30 | 92.81±0.10 | 93.52±0.31 | 94.51±0.20 | 94.48±0.24$^t$ | 94.63±0.23$^t$ | **94.77±0.13**$^s$ |
| ResNeXt-A | 94.99 | 93.39±0.67 | 94.05±0.48 | 94.24± 0.22 | 94.25±0.21$^t$ | 94.12±0.20$^t$ | **94.37±0.14**$^t$ |
| ResNeXt-B | 95.12 | 93.56±0.33 | 93.65±0.64 | 94.33±0.26 | **94.29±0.24**$^t$ | 94.26±0.35$^t$ | 94.23±0.32$^t$ |

In the following experiments, we evaluate three searched proxy tasks, denoted by **GenNAS search-N**, **-D**, and **-R**, meaning that the task is searched on NASBench-101, NDS DARTS search space, and NDS ResNet search space, respectively. We study the performance and transferability of the searched tasks on all NAS search spaces. Proxy task search is done on a single GPU GTX 1080Ti. On NASBench-101 (**GenNAS-N**), ResNeXt (**GenNAS-R**), and DARTS (**GenNAS-D**), the search time is 5.75, 4, and 12.25 GPU hours, respectively. Once the proxy tasks is searched, it can be used to evaluate any architectures in the target search space and can be transferred to other search spaces.

**GenNAS with Searched/Transferred Proxy Task**. The performance of GenNAS-search-N, -D, and -R proxy tasks is shown in Table 2. First, in most cases, proxy task search improves the ranking correlation. For example, in NDS, using the proxy task searched on DARTS space (search-D) outperforms other GenNAS settings on DARTS-like spaces, while using proxy task search-R on ResNet-like spaces outperforms others as well. In NASBench-NLP, the proxy task search can push the ranking correlation to 0.81, surpassing the ppl@ep3 (0.79). Such results demonstrate the effectiveness of our proposed proxy task search. Second, the searched proxy task shows great transferability: the proxy task searched on NASBench-101 (search-N) generally works well for other search spaces, i.e., NASBench-201, NDS, and NASBench-NLP. This further emphasizes that the fundamental principles for top-performing neural architectures are similar across different tasks and datasets. Fig. 5 visualizes the performance of GenNAS comparing with others.

## 4.4 GenNAS for End-to-End NAS

Finally, we evaluate GenNAS on the end-to-end NAS tasks, aiming to find the best neural architectures within the search space. Table 3 summarizes the comparisons with the state-of-the-art NAS approaches, including previously used NASWOT, synflow, cls@ep10, and additionally Halfway [30], RSPS [28], DARTS-V1 [3], DARTS-V2, GDAS [58], SETN [59], and ENAS [60]. Halfway is the NASBench-101-released result using half of the total epochs for network training. In all the searches during NAS, we do not use any tricks such as warmup selection [31] or groundtruth query to compensate the low rank correlations. We fairly use a simple yet effective regularized evolutionary algorithm [1] and adopt the proposed regression loss as the fitness function. The population size is 50 and the tournament sample size is 10 with 400 iterations. On NASBench-101, GenNAS finds better architectures than NASWOT and Halfway while being up to $200\times$ faster. On NASBench-201, GenNAS finds better architectures than the state-of-the-art GDAS within 0.3 GPU hours, being $30\times$ faster. Note that GenNAS uses the proxy task searched on NASBench-101 and transferred to NASBench-201, demonstrating remarkable transferability. On Neural Design Spaces, GenNAS finds better architectures than the cls@ep10 using labeled classification while being $40\times$ faster. On NASBench-NLP, GenNAS finds architectures that achieve 0.246 (the lower the better) average final regret score $r$, outperforming the ppl@ep3 (0.268) with $192\times$ speed up. The regret score $r$ at the

Table 4: Comparisons with state-of-the-art NAS methods on ImageNet under the mobile setting. $^*$ is the time for proxy task search.

| Method | Test Err. (%) top-1 | Test Err. (%) top-5 | Params (M) | FLOPS(M) (M) | Search Cost (GPU days) | Searched Method | Searched dataset |
|---|---|---|---|---|---|---|---|
| NASNet-A [18] | 26.0 | 8.4 | 5.3 | 564 | 2000 | RL | CIFAR-10 |
| AmoebaNet-C [1] | 24.3 | 7.6 | 6.4 | 570 | 3150 | evolution | CIFAR-10 |
| PNAS [63] | 25.8 | 8.1 | 5.1 | 588 | 225 | SMBO | CIFAR-10 |
| DARTS(2nd order) [3] | 26.7 | 8.7 | 4.7 | 574 | 4.0 | gradient-based | CIFAR-10 |
| SNAS [64] | 27.3 | 9.2 | 4.3 | 522 | 1.5 | gradient-based | CIFAR-10 |
| GDAS [58] | 26.0 | 8.5 | 5.3 | 581 | 0.21 | gradient-based | CIFAR-10 |
| P-DARTS [65] | 24.4 | 7.4 | 4.9 | 557 | 0.3 | gradient-based | CIFAR-10 |
| P-DARTS | 24.7 | 7.5 | 5.1 | 577 | 0.3 | gradient-based | CIFAR-100 |
| PC-DARTS [32] | 25.1 | 7.8 | 5.3 | 586 | 0.1 | gradient-based | CIFAR-10 |
| TE-NAS [62] | 26.2 | 8.3 | 6.3 | - | 0.05 | training-free | CIFAR-10 |
| PC-DARTS | 24.2 | 7.3 | 5.3 | 597 | 3.8 | gradient-based | ImageNet |
| ProxylessNAS [4] | 24.9 | 7.5 | 7.1 | 465 | 8.3 | gradient-based | ImageNet |
| UNNAS-jig [43] | 24.1 | - | 5.2 | 567 | 2 | gradient-based | ImageNet |
| TE-NAS | 24.5 | 7.5 | 5.4 | 599 | 0.17 | training-free | ImageNet |
| **GenNAS-combo** | 25.1 | 7.8 | 4.8 | 559 | 0.04 | evolution+few-shot | CIFAR-10 |
| **GenNAS-D14** | 24.3 | 7.2 | 5.3 | 599 | $0.7^*$+0.04 | evolution+few-shot | CIFAR-10 |

moment $t$ is $r(t) = L(t) - L^\star$, where $L(t)$ is the testing log perplexity of the best found architecture according to the prediction by the moment, and $L^\star = 4.36$ is the lowest testing log perplexity in the whole dataset achieved by LSTM [50] architecture.

On DARTS search space, we also perform the end-to-end search on ImageNet-1k [61] dataset. We fix the depth (layer) of the networks to be 14 and adjust the width (channel) so that the # of FLOPs is between 500M to 600M. We evaluate two strategies: one without task search using GenNAS-combo (see Table 1), and the other GenNAS-D14 with proxy task search on DARTS search space with depth 14 and initial channel 16. The training settings follow TENAS [62]. The results are shown in Table 4. We achieve top-1/5 test error of 25.1/7.8 using GenNAS-combo and top-1/5 test error of 24.3/7.2 using GenNAS-D14, which are on par with existing NAS architectures. GenNAS-combo is much faster than existing works, while GenNAS-D14 pays extra search time cost. Our next step is to investigate the searched tasks and demonstrate the generalization and transferrability of those searched tasks to further reduce the extra search time cost.

These end-to-end NAS experiments strongly suggest that GenNAS is generically efficient and effective across different search spaces and datasets.

## 5  Conclusion

In this work, we proposed GenNAS, a self-supervised regression-based approach for neural architecture training, evaluation, and search. GenNAS successfully answered the two questions at the beginning. (1) GenNAS *is* a generic task-agnostic method, using synthetic signals to capture neural networks' fundamental learning capability without specific downstream task knowledge. (2) GenNAS *is* an extremely effective and efficient method using regression, fully utilizing all the training samples and better capturing valued information. We show the true power of self-supervised regression via manually designed proxy tasks that do not need to search. With proxy search, GenNAS can deliver even better results. Extensive experiments confirmed that GenNAS is able to deliver state-of-the-art performance with near-zero search time, in terms of both ranking correlation and the end-to-end NAS tasks with great generalizability.

## 6  Acknowledgement

We thank IBM-Illinois Center for Cognitive Computing Systems Research (C3SR) for supporting this research. We thank all reviewers and the area chair for valuable discussions and feedback. This work utilizes resources [66] supported by the National Science Foundation's Major Research Instrumentation program, grant #1725729, as well as the University of Illinois at Urbana-Champaign. P.L. is also partly supported by the 2021 JP Morgan Faculty Award and the National Science Foundation award HDR-2117997.

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
