# A Appendix

## A.1 NAS Search Spaces

NASBench-101[1] introduces a large and expressive search space with 423k unique convolutional neural architectures and training statistics on CIFAR-10. NASBench-201[2] contains the training statistics of 15,625 architectures across three different datasets, including CIFAR-10, CIFAR-100, and Tiny-ImageNet-16. Network Design Spaces (NDS) dataset[3] [1] with PYCLS [2] codebase [4] provides trained neural networks from 11 search spaces including DARTS [3], AmoebaNet [4], ENAS [5], NASNet [6], PNAS [7], ResNet [8], ResNeXt [9], etc. A search space with "-f" suffix stands for a search space that has fixed number of layers and channels. The ResNeXt-A and ResNeXt-B have different channel-number and group-convolution settings. NASBench-NLP[5] [10] is an NLP neural architecture search space, including 14k recurrent cells trained on the Penn Treebank (PTB) [11] dataset.

## A.2 Experiment Setup

For **CNN** architecture training, the learning rate is $1e-1$ and the weight decay is $1e-5$. Each architecture is trained for 100 iterations on a single batch of data using SGD optimizer [12]. All the convolutional architectures use the same setting. The size of the input images is $h = w = 32$ and the size of output feature maps is $h' = w' = 8$. For **RNN** training, the learning rate is $1e-3$, the weight decay is $1.2e-6$, the batch size $b$ is 16, and the sequence length $l$ is 8. Each architecture is trained for 100 iterations on a single batch of data using the Adam optimizer [13].

### A.2.1 Regularized Evolutionary Algorithm

---
**Algorithm 1** Regularized Evolutionary Algorithm in General

---
Initialize an empty population queue, $Q\_pop$ // The maximum population is $P$
Initialize an empty set, $history$ // Will contain all visited individuals
**for** $i = 1, 2, \cdots, P$ **do**
    $new\_individual \leftarrow$ RandomInit()
    $new\_individual.fitness \leftarrow$ Eval($new\_individual$)
    Enqueue($Q\_pop, new\_individual$)// Add individual to the right of $Q\_pop$
    Add $new\_individual$ to $history$
**end**
// Evolve for $T\_iter$
**for** $i = 1, 2, \cdots, T\_iter$ **do**
    Initialize an empty set, $sample\_set$
    **for** $i = 1, 2, \cdots, S$ **do**
        Add an individual to $sample\_set$ from $Q\_pop$ without replacement.
        // The individual stays in $Q\_pop$
    **end**
    $parent \leftarrow$ the individual with best fitness in $sample\_set$
    $child \leftarrow$ Mutate($parent$)
    $child.fitness \leftarrow$ Eval($child$)
    Enqueue( $Q\_pop, child$ )
    Add $child$ to $history$
    Dequeue( $Q\_pop$)// Remove the oldest individual from the left of $Q\_pop$
**end**
**return** the individual with best fitness in $history$

---

Regularized Evolutionary Algorithm [4] (RE) combines the tournament selection [14] with the aging mechanism which remove the oldest individuals from the population each round. We show a general form of RE in Alg. 1. Aging evolution aims to explore the search space more extensively, instead of

---

[1] https://github.com/google-research/nasbench
[2] https://github.com/D-X-Y/NAS-Bench-201
[3] https://github.com/facebookresearch/nds
[4] https://github.com/facebookresearch/pycls
[5] https://github.com/fmsnew/nas-bench-nlp-release

focusing on the good models too early. Works [15, 16, 17] also suggest that the RE is suitable for neural architecture search. Since we aim to develop a general NAS evaluator (as the fitness function in RE), we conduct fair comparisons between GenNAS and other methods without fine-tuning or any tricks (e.g., warming-up). Hence, we constantly use the setting $P = 50$, $S = 10$, $T\_iter = 400$ for all the search experiments.

### A.2.2 Proxy Task Search

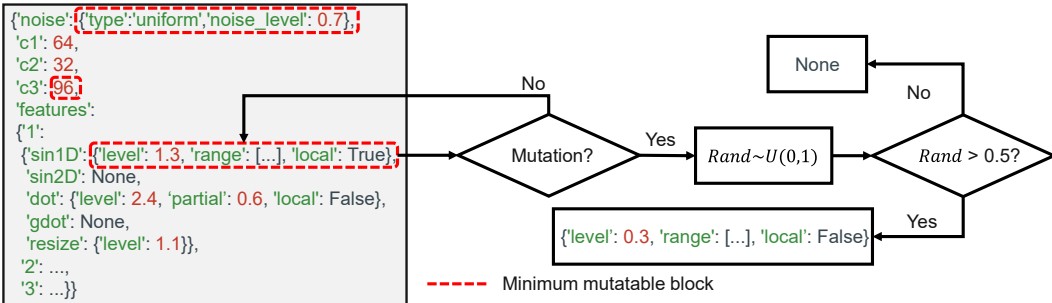

Figure 1: The configuration of a task in JSON style and the illustration of task mutation.

The configuration of a task is shown in Fig. 1. We introduce the detailed settings of different signal bases. (1) Noise is chosen from standard Gaussian distribution ($\mathcal{N}(0,1)$) or uniform distribution ($\mathcal{U}(-1,1)$). The generated noise maps are directly multiplied by the level which can be selected from 0 to 1 with a step of 0.1. (2) Sin1D generates 2D synthetic feature maps using different frequencies choosing from the range, which contains 10 frequencies sampled from $[a, b]$, where a and b are sampled from 0 to 0.5 with the constraint $0 < a < b < 0.5$. (3) Sin2D uses the similar setting as Sin1D, where both the $f_x$ and $f_y$ for a 2D feature map are sampled from the range. (4) $C_i$ can be selected from $\{16, 32, 48, 64, 96\}$. Other settings are already described in Section 3.1.2. During the mutation, each minimum mutatable block (including signal definitions and the number of channels) has 0.2 probability to be regenerated as shown in Fig. 1. For RNN settings, we search for both the input and output synthetic signal tensors. The dimension $d$ is chosen from $\{16, 32, 48, 64, 96\}$.

### A.2.3 End-to-end NAS

In the end-to-end NAS, GenNAS is incorporated in the RE as the fitness function to explore the search space. For **NASBench-101**, the mutation rate for each vertice is $1/|v|$ where $|v| = 7$. More details of the search space NASBench-101 can be found in the original paper [16]. For **NASBench-201**, the mutation rate for each edge is $1/|e|$ where $|e| = 6$. More details of the search space NASBench-101 can be found in the original paper [15]. For **NDS** ResNet series, the sub search space consists of 25000 sampled architectures from the whole search space. We apply mutation in RE by randomly sampling 200 architectures from the sub search space and choosing the most similar architecture as the child. For **NASBench-NLP**, we follow the work [10] by using the graph2vec [18] features find the most similar architecture as the child.

### A.3 Additional Experiments

### A.3.1 Regression vs. Classification Using Same Training Samples

We study effectiveness of regression using 10 tasks searched on NASBench-101, varying the batch size from 1 to 1024. The ranking correlation achieved by GenNAS using regression is plotted in Fig. 2a. We also plot the classification task performances with single-batch data in Fig. 2b. Apparently, using the same amount of data, the ranking correlation achieved by classification ($\rho$ around 0.85) is much worse than regression ($\rho$ around 0.3).

### A.3.2 Batch Sizes and Iterations

We show the effect of batch size by using 10 tasks searched on NASBench-101. We use batch sizes varied from 1 to 1024 and plot the ranking results in Fig. 2a. We find that 16 as the batch

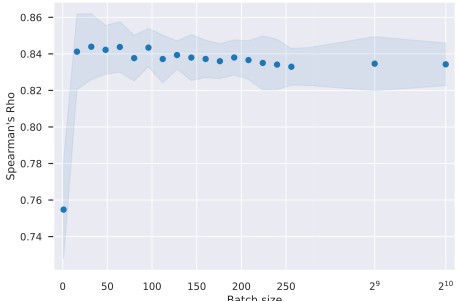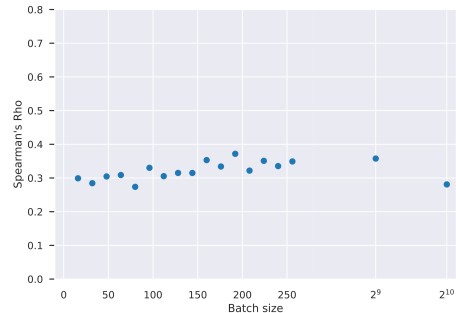

(a) Regression (GenNAS) ranking correlation averaged from 10 searched tasks using different batch sizes.

(b) Classification task's ranking correlation using the same setting as (a).

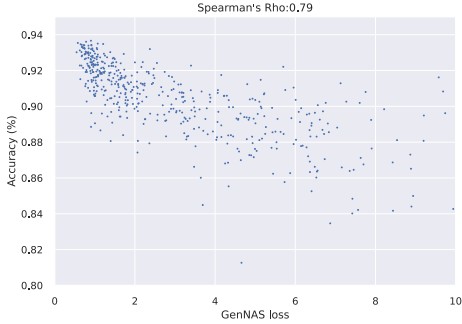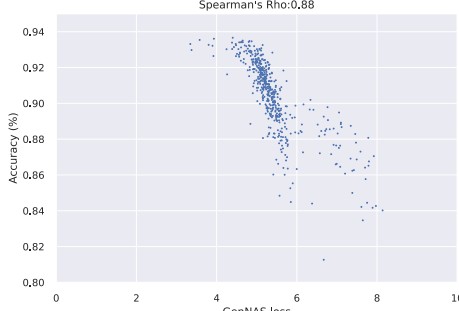

(c) Classification accuracy vs. GenNS regression loss with batch size 1.

(d) Classification accuracy vs. GenNS regression loss with batch size 16.

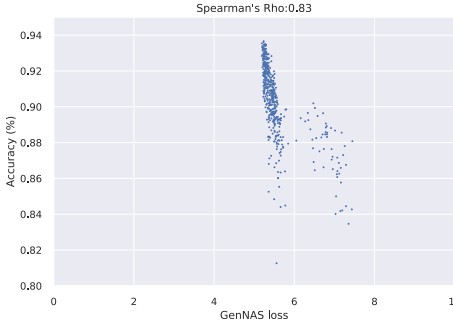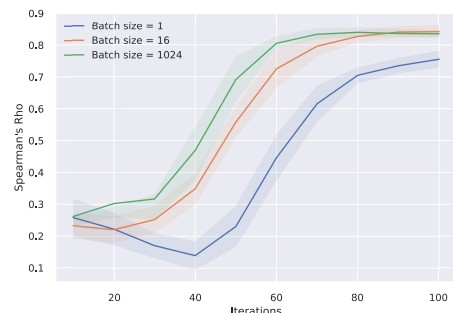

(e) Classification accuracy vs. GenNS regression loss with batch size 1024.

(f) 10 searched tasks' ranking correlation on NASBench-101 using different numbers of iterations.

Figure 2: (a) Regression (GenNAS) ranking correlation averaged from 10 searched tasks on NASBench-101 in terms of Spearman's $\rho$ with batch size in $\{1, 16, 32, ..., 256, 512, 1024\}$. (b) Classification task's ranking correlation using the same amount of data in (a). (c) Classification accuracy using a searched task on NASBench-101 with batch size as 1 on CIFAR-10. The y-axis is the groundtruth (CIFAR-10 accuracy) and the x-axis is the GenNAS regression loss. (d) Similar to (c), batch size as 16. (e) Similar to (c), batch size as 1024. (f) 10 searched tasks ranking performance (in terms of Spearman's $\rho$) on NASBench-101 using different iterations.

size is adequate for a good ranking performance. Also, we observe a small degradation when batch size increases and then becomes stable as the batch size keeps increasing. Hence, We plot the ranked architecture distribution with a searched task using 1, 16, 1024 as batch size respectively on Fig. 2c, 2d, 2e. We observe that when only using a single image, the poor-performance architectures can also achieve similar regression losses as the good architectures. It suggests that the task is too easy to distinguish the differences among architectures. Also, 1024 as batch size leads to the higher regression losses of best architectures. It suggests that a very challenging task is also hard for good

architectures to learn and may lead to slight ranking performance degradation. In addition, we plot the ranking performance using different numbers of iterations in Fig. 2f. It shows that 100 iterations is necessary for the convergence of ranking performance.

### A.3.3 Sensitivity Studies of Random Seeds and Initialization

Table 1: Ranking correlation (Spearman's $\rho$) analysis of different 10 seeds across three different search spaces with the searched tasks on them respectively. For the NASBench-101, the 500 architecture samples [19] are constantly used for evaluation. For DARTS and ResNet search spaces, 1000 samples are randomly sampled with different seeds from the evaluated architecture sets provided by NDS [1].

| Search Space | 0 | 1 | 2 | 3 | 4 | 5 | 6 | 7 | 8 | 9 | Average |
|---|---|---|---|---|---|---|---|---|---|---|---|
| NASBench-101 | 0.880 | 0.850 | 0.875 | 0.869 | 0.872 | 0.874 | 0.877 | 0.863 | 0.872 | 0.872 | 0.870±0.008 |
| DARTS | 0.809 | 0.899 | 0.861 | 0.831 | 0.841 | 0.836 | 0.851 | 0.841 | 0.885 | 0.861 | 0.850±0.025 |
| ResNet | 0.860 | 0.853 | 0.841 | 0.810 | 0.865 | 0.877 | 0.874 | 0.804 | 0.808 | 0.803 | 0.840±0.029 |

**Random Seeds.** We rerun the 3 searched tasks on their target search spaces (NASBench-101, DARTS, ResNet) for 10 runs with different random seeds. The results are shown in Table 1. GenNAS demonstrates its robustness across different random seeds.

Table 2: Ranking correlation (Spearman's $\rho$) analysis of 10 searched tasks on NASBench-101 with different initialization methods.

| Weight init | Bias init | 0 | 1 | 2 | 3 | 4 | 5 | 6 | 7 | 8 | 9 | Average |
|---|---|---|---|---|---|---|---|---|---|---|---|---|
| Default | Default | 0.835 | 0.860 | 0.860 | 0.878 | 0.835 | 0.810 | 0.859 | 0.832 | 0.816 | 0.828 | 0.841±0.021 |
| Kaiming | Default | 0.844 | 0.854 | 0.857 | 0.856 | 0.832 | 0.818 | 0.854 | 0.746 | 0.829 | 0.811 | 0.830±0.032 |
| Xavier | Default | 0.856 | 0.881 | 0.863 | 0.874 | 0.849 | 0.825 | 0.865 | 0.830 | 0.838 | 0.851 | 0.853±0.018 |
| Default | Zero | 0.867 | 0.882 | 0.854 | 0.880 | 0.848 | 0.847 | 0.874 | 0.808 | 0.848 | 0.850 | 0.856±0.021 |
| Kaiming | Zero | 0.845 | 0.842 | 0.856 | 0.861 | 0.828 | 0.821 | 0.846 | 0.770 | 0.823 | 0.823 | 0.831±0.025 |
| Xavier | Zero | 0.859 | 0.876 | 0.869 | 0.879 | 0.839 | 0.842 | 0.861 | 0.828 | 0.846 | 0.843 | 0.854±0.016 |

**Initialization.** We perform an experiment to evaluate the effects of different initialization for 10 searched tasks on NASBench-101. For the weights, we use the default PyTorch initialization, Kaiming initialization [20], and Xavier initialization [21]. For the bias, we use the default PyTorch initialization and zeroized initialization. The results are shown in Table 2. We observe that for some specific tasks (e.g., task 7), Kaiming initialization may lead to lower ranking correlation. Also, zeroized bias initialization slightly increases the ranking correlation. However, overall, GenNAS shows stable performance across different initialization methods.

### A.3.4 Kendall Tau and Retrieving Rates

For the sample experiments on NASBench-101/201/NLP and NDS, we report the performance of our methods compared to other efficient NAS approaches' in Table 3 by Kendall $\tau$ [22]. We define the retrieving rate@topK as $\frac{\#\{\text{Pred@TopK} \cap \text{GT@TopK}\}}{\#\{\text{GT@TopK}\}}$, where $\#$ is the operator of cardinality, GT@TopK and Pred@TopK are the set of architectures that are ranked in the top-K of groundtruths and predictions respectively. We report the retrieving rate@Top10% for all the search spaces in Table 4. Moreover, we report the retrieving rate@Top5%-Top50% for GenNAS-COMBO and GenNAS-N on NASBench-101 with other 1000 random sampled architectures in Table 5.

### A.3.5 End-to-end NAS Architectures

Here we visualize all the ImageNet DARTS cell architectures: searched by GenNAS-combo, searched by GenNAS-D14 in Fig 3 and Fig 4 respectively.

### A.3.6 GPU Performance

We use the PyTorch 1.5.0 [26], on a desktop with I7-6700K CPU, 16 GB RAM and a GTX 1080 Ti GPU (11GB GDDR5X memory) to evaluate the GPU performance of GenNAS. The results are shown in Table 6.

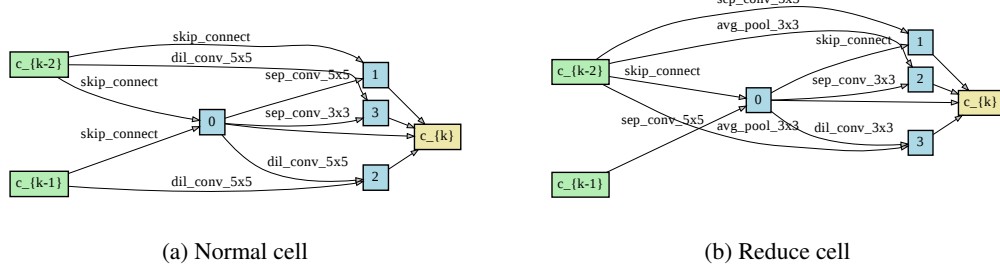

| (a) Normal cell | (b) Reduce cell |

Figure 3: Cell architectures (normal and reduce) searched by GenNAS-combo

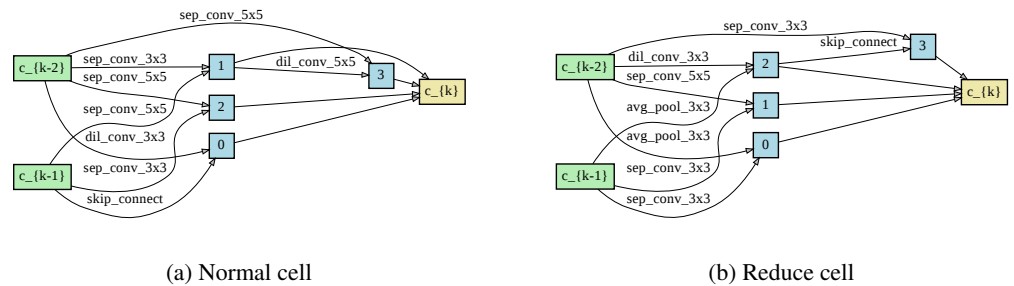

| (a) Normal cell | (b) Reduce cell |

Figure 4: Cell architectures (normal and reduce) searched by GenNAS-D14

Table 3: GenNAS' ranking correlation evaluation comparing with other efficient NAS approaches using the Kendall $\tau$.

**NASBench-101**

| Dataset | NASWOT [23] | synflow [24] | GenNAS | | |
|---|---|---|---|---|---|
| | | | single | combo | search-N |
| CIFAR-10 | 0.27 | 0.24 | 0.59 | 0.66 | **0.7** |
| ImgNet | 0.36 | 0.14 | 0.47 | **0.54** | 0.53 |

**NASBench-201**

| Dataset | NASWOT | synflow | jacob_cov | snip | EcoNAS [25] | GenNAS | | |
|---|---|---|---|---|---|---|---|---|
| | | | | | | single | combo | search-N |
| CIFAR-10 | 0.6 | 0.52 | 0.59 | 0.41 | 0.62 | 0.57 | 0.67 | **0.71** |
| CIFAR-100 | 0.63 | 0.57 | 0.53 | 0.46 | 0.57 | 0.52 | 0.63 | **0.65** |
| ImgNet16 | 0.6 | 0.54 | 0.56 | 0.44 | 0.57 | 0.53 | 0.61 | **0.67** |

**Neural Design Spaces**

| Dataset | NAS-Space | NASWOT | synflow | GenNAS | | | | |
|---|---|---|---|---|---|---|---|---|
| | | | | single | combo | search-N | search-D | search-R |
| CIFAR-10 | DARTS | 0.48 | 0.3 | 0.3 | 0.45 | 0.52 | **0.68** | 0.63 |
| | DARTS-f | 0.21 | 0.09 | 0.36 | **0.43** | 0.37 | 0.42 | 0.36 |
| | Amoeba | 0.21 | 0.06 | 0.36 | 0.47 | 0.5 | **0.59** | 0.53 |
| | ENAS | 0.39 | 0.13 | 0.39 | 0.49 | 0.48 | **0.63** | 0.59 |
| | ENAS-f | 0.31 | 0.2 | 0.46 | **0.55** | 0.49 | 0.53 | 0.48 |
| | NASNet | 0.3 | 0.02 | 0.4 | 0.5 | 0.47 | **0.58** | 0.52 |
| | PNAS | 0.36 | 0.17 | 0.22 | 0.37 | 0.42 | **0.57** | 0.52 |
| | PNAS-f | 0.09 | 0.18 | 0.31 | 0.38 | 0.39 | **0.38** | 0.33 |
| | ResNet | 0.19 | 0.14 | 0.23 | 0.38 | 0.38 | 0.38 | **0.64** |
| | ResNeXt-A | 0.46 | 0.32 | 0.4 | 0.5 | 0.6 | 0.45 | **0.65** |
| | ResNeXt-B | 0.4 | 0.43 | 0.17 | 0.3 | 0.37 | 0.38 | **0.52** |
| ImageNet | DARTS | 0.49 | 0.14 | 0.43 | 0.52 | 0.52 | **0.66** | 0.48 |
| | DARTS-f | 0.13 | 0.25 | 0.49 | **0.57** | 0.48 | 0.51 | 0.42 |
| | Amoeba | 0.33 | 0.17 | 0.46 | 0.53 | 0.55 | **0.62** | 0.5 |
| | ENAS | 0.51 | 0.12 | 0.4 | 0.47 | 0.4 | **0.63** | 0.48 |
| | NASNet | 0.39 | 0.01 | 0.36 | 0.42 | 0.37 | **0.5** | 0.43 |
| | PNAS | 0.45 | 0.11 | 0.19 | 0.27 | 0.31 | **0.45** | 0.3 |
| | ResNeXt-A | 0.52 | 0.28 | 0.61 | **0.7** | 0.56 | 0.44 | 0.69 |
| | ResNeXt-B | 0.45 | 0.21 | 0.53 | 0.65 | 0.39 | 0.43 | **0.67** |

**NASBench-NLP**

| Dataset | GenNAS | | |
|---|---|---|---|
| | single | combo | search |
| PTB | 0.43 | 0.55 | 0.63 |

Table 4: GenNAS' retrieving rate@top10% comparing with other efficient NAS approaches. For the NASBench-101 we use the set of 500 architectures that sampled by Liu, et al. [19] for obtaining the ImageNet groundtruth.

**NASBench-101**

| Dataset | number of samples | NASWOT [23] | synflow [24] | GenNAS single | GenNAS combo | GenNAS search-N |
|---------|-------------------|-------------|--------------|---------------|--------------|-----------------|
| CIFAR-10 | 500 | 32% | 28% | 58% | 64% | **68%** |
| ImgNet | 500 | 36% | 14% | 52% | 54% | **64%** |

**NASBench-201**

| Dataset | number of samples | NASWOT | synflow | jacob_cov | snip | EcoNAS [25] | GenNAS single | GenNAS combo | GenNAS search-N |
|---------|-------------------|--------|---------|-----------|------|-------------|---------------|--------------|-----------------|
| CIFAR-10 | 1000 | 43% | 48% | 27% | 27% | 52% | 43% | 36% | **53%** |
| CIFAR-100 | 1000 | 48% | 47% | 23% | 36% | 47% | 46% | 46% | **58%** |
| ImgNet16 | 1000 | 49% | 43% | 33% | 32% | 41% | 48% | 40% | **51%** |

**Neural Design Spaces**

| Dataset | number of samples | NAS-Space | NASWOT | synflow | GenNAS single | GenNAS combo | GenNAS search-N | GenNAS search-D | GenNAS search-R |
|---------|-------------------|-----------|--------|---------|---------------|--------------|-----------------|-----------------|-----------------|
| CIFAR-10 | 1000 | DARTS | 29% | 10% | 16% | 43% | 45% | **59%** | 49% |
| | 1000 | DARTS-f | 1% | 5% | 22% | **33%** | 18% | 22% | 23% |
| | 1000 | Amoeba | 20% | 4% | 20% | 39% | 45% | **50%** | 40% |
| | 1000 | ENAS | 31% | 6% | 25% | 48% | 41% | **57%** | 48% |
| | 1000 | ENAS-f | 28% | 2% | 34% | **45%** | 42% | 38% | 37% |
| | 1000 | NASNet | 33% | 7% | 27% | 38% | **46%** | 52% | 43% |
| | 1000 | PNAS | 24% | 9% | 21% | 39% | **46%** | 44% | 37% |
| | 1000 | PNAS-f | 6% | 4% | 21% | 27% | **31%** | 25% | 22% |
| | 1000 | ResNet | 7% | 4% | 38% | 44% | 38% | 54% | **64%** |
| | 1000 | ResNeXt-A | 28% | 25% | 25% | **61%** | 53% | 52% | 58% |
| | 1000 | ResNeXt-B | 21% | 30% | 10% | 36% | 13% | 40% | **71%** |
| ImageNet | 121 | DARTS | 17% | 0% | 50% | 58% | 55% | **58%** | 18% |
| | 499 | DARTS-f | 8% | 4% | 33% | 27% | 35% | **39%** | 24% |
| | 124 | Amoeba | 33% | 0% | 50% | 42% | 58% | **58%** | 41% |
| | 117 | ENAS | 36% | 9% | 18% | 18% | 45% | **55%** | 45% |
| | 122 | NASNet | 33% | 0% | 42% | **50%** | 42% | 33% | 33% |
| | 119 | PNAS | 10% | 9% | 45% | 36% | 45% | **55%** | 9% |
| | 130 | ResNeXt-A | 31% | 8% | 67% | 67% | 50% | 33% | **75%** |
| | 164 | ResNeXt-B | 38% | 13% | 38% | 50% | 33% | 38% | **64%** |

**NASBench-NLP**

| Dataset | number of samples | grad | norm | snip | grasp | fisher | synflow | GenNAS single | GenNAS combo | GenNAS search |
|---------|-------------------|------|------|------|-------|--------|---------|---------------|--------------|---------------|
| PTB | 1000 | 10% | 10% | 4% | - | 22% | 38% | 38% | 47% | **63%** |

Table 5: Retrieving rate@top5%-top50% of GenNAS-combo/N on 1000 randomly sampled architectures on NASBench-101.

| Method | @top5% | @top10% | @top20% | @top30% | @top40% | @top50% |
|--------|--------|---------|---------|---------|---------|---------|
| GenNAS-combo | 0.6 | 0.6 | 0.73 | 0.74 | 0.79 | 0.82 |
| GenNAS-N | 0.56 | 0.58 | 0.66 | 0.76 | 0.80 | 0.85 |

Table 6: Evaluations of GenNAS' GPU performance. We test GenNAS with 6 different batch sizes from 16 to 96. "A/B" denotes: A (second) as the average one-iteration run time for the search space, and B (GB or gigabyte) as the GPU memory usage. "OOM" means some large models may lead to the out-of-memory issue for the target GPU.

| Search Space | B-size 16 | B-size 32 | B-size 48 | B-size 64 | B-size 80 | B-size 96 |
|--------------|-----------|-----------|-----------|-----------|-----------|-----------|
| NASBench-101 | 0.023/0.78 | 0.023/0.92 | 0.023/1.12 | 0.022/1.22 | 0.023/1.41 | 0.023/1.56 |
| NASBench-201 | 0.020/0.77 | 0.020/0.93 | 0.021/1.12 | 0.020/1.24 | 0.020/1.46 | 0.020/1.62 |
| DARTS | 0.049/1.92 | 0.056/3.39 | 0.069/5.07 | 0.088/6.88 | 0.103/7.62 | OOM |
| DARTS-f | 0.077/1.42 | 0.088/2.21 | 0.104/3.30 | 0.122/3.79 | 0.145/5.15 | 0.171/6.07 |
| Amoeba | 0.080/2.53 | 0.103/4.38 | 0.128/6.60 | 0.159/9.25 | 0.194/6.85 | OOM |
| ENAS | 0.059/2.40 | 0.076/4.31 | 0.090/5.84 | 0.115/7.78 | 0.140/9.15 | OOM |
| ENAS-f | 0.095/1.67 | 0.111/2.58 | 0.134/3.70 | 0.159/4.54 | 0.186/6.31 | 0.216/7.53 |
| NASNet | 0.061/2.23 | 0.073/3.77 | 0.094/5.83 | 0.116/6.87 | 0.140/8.68 | 0.160/9.30 |
| PNAS | 0.074/2.52 | 0.097/4.36 | 0.121/6.53 | 0.155/8.23 | 0.189/9.42 | OOM |
| PNAS-f | 0.114/1.69 | 0.143/2.67 | 0.173/4.03 | 0.208/4.83 | 0.250/6.24 | 0.293/7.45 |
| ResNet | 0.016/1.28 | 0.016/1.54 | 0.016/2.05 | 0.016/2.34 | 0.016/2.34 | 0.016/2.77 |
| ResNeXt-A | 0.025/1.65 | 0.025/2.82 | 0.026/4.55 | 0.027/6.86 | 0.028/9.75 | 0.029/5.92 |
| ResNeXt-B | 0.022/1.98 | 0.022/3.47 | 0.023/5.63 | 0.024/7.33 | 0.027/9.51 | 0.029/5.95 |
| NASBench-NLP | 0.029/0.90 | 0.029/0.90 | 0.029/0.90 | 0.029/0.90 | 0.029/0.93 | 0.029/0.94 |