# OpenReview forum: "Generic Neural Architecture Search via Regression"
_NeurIPS.cc/2021/Conference — NeurIPS 2021 Spotlight_

### Official Review · Reviewer_CCjK · 2021-07-12

**Rating:** 7
**Confidence:** 5

**Summary:**

This work designed synthetic signals as proxy tasks for Neural Architecture Search. The author claimed that self-supervised regression tasks on the synthetic signals can deliver task-agnositc NAS, and make searched results more highly correlated with ground truth accuracy of architectures.

**Limitations And Societal Impact:**

My main concern is the usage of the regression tasks on manually designed signals, which may not widely adoptable (than unlabeld natural images) and further cost design efforts.

**Main Review:**

1. "thus" in the first item in the contribution list on page 2: I don't think it is the "switching from classification to regression in NAS" that leads to being "agnostic to the speciﬁc downstream tasks". This description could be misleading.
2. Making NAS agnostic to tasks is a reasonable motivation, but this is irrelevant using regression or classification as the loss function, but more related to whether or not using supervision signals. Many places (e.g. line 56-57, 139) make a reader feel that regression is vital, but I disagree. Introducing regression tasks in NAS is not a novelty. Properly designing proxy tasks is the root.
3. I could not see the necessity of using manually designed signals. If this is true, we should use large amount of manually designed signals for contrastive learning, to learn meaningful representations before downstream fine-tuning. Actually self-supervised learning on large-scale datasets like ImageNet, JFT can work very well recently [1,2]. Not to say the effort and uncertainty of manually designing these signals.
4. I think the correlation comparison in Table 2 is not fair. GenNAS can use combo of signals, but all other methods only measure the correlation on a single dataset. Maybe using combination of CIFAR and ImageNet-16 can further boost their correlation.
5. Another efficient & w.o. proxy task NAS work is not considered [3].

[1] Improved Baselines with Momentum Contrastive Learning

[2] A simple framework for contrastive learning of visual representations

[3] Neural Architecture Search on ImageNet in Four GPU Hours: A Theoretically Inspired Perspective

**Time Spent Reviewing:**

1.5 hour

---

> ### Author Response · Authors · 2021-08-10
> **Replies to CCjK**
>
> We thank the reviewer for the very detailed and useful comments. Here we would like to address the reviewer’s questions:
>
> Q: "thus" in the first item in the contribution list on page 2: I don't think it is the "switching from classification to regression in NAS" that leads to being "agnostic to the speciﬁc downstream tasks". This description could be misleading.
>
> Making NAS agnostic to tasks is a reasonable motivation, but this is irrelevant using regression or classification as the loss function, but more related to whether or not using supervision signals. Many places (e.g. line 56-57, 139) make a reader feel that regression is vital, but I disagree. Introducing regression tasks in NAS is not a novelty. Properly designing proxy tasks is the root.
>
> A: First, we appreciate the reviewer’s comment on task-agnostic, which we also believe is important and will run more experiments to support this claim in the future. In this paper, we claim it is “task-agnostic” because of three reasons: (1) in both CV and NLP models, the supervising task is a regression task but the target task is classification; (2) the general idea of GenNAS, i.e., using synthetic signals in regression, works well for both CV and NLP tasks; (3) for the classification tasks in CV, the datasets and search spaces are very diverse, but our GenNAS is dataset-agnostic: the searched network architecture on one dataset naturally works well on other datasets. Such features of GenNAS makes it highly adaptive to unseen tasks/datasets.
>
> Second, we would like to explain more about the importance of using regression. We believe we are the very first work that explicitly proposes regression as the self-supervised task, and we believe that introducing regression in NAS is a key contribution because of three reasons: (1) exactly as the reviewer mentioned, designing the proxy tasks, i.e., supervision signals, is the root. Using regression as the proxy task allows us to better design and select these important supervision signals and to evaluate the capability of the network architectures to capture the supervision signals. If using classification as the proxy task, there is no direct way to do it. (2) as illustrated in Fig. 1 and shown in the experiments, using regression significantly shortens the training time (by orders of magnitude), reduces the required training sample (only one batch of image), and maintains high accuracy (much higher ranking correlation score compared with zero-training methods). (3) using regression as the proxy task can fully utilize valuable and rich intermediate information (e.g., hidden layers’ outputs) other than only using the last layer’s outputs (e.g., the classifier’s output), which leads to much faster convergence and can better evaluate a network architecture’s intrinsic quality; In appendix, Figure 2 a&b indeed suggest that our method achieves better results compared to the classification with the same amount of data with respect to ranking correlation (~0.8 v.s. ~0.4).
>
>
> Q: I could not see the necessity of using manually designed signals. If this is true, we should use large amount of manually designed signals for contrastive learning, to learn meaningful representations before downstream fine-tuning. Actually self-supervised learning on large-scale datasets like ImageNet, JFT can work very well recently. Not to say the effort and uncertainty of manually designing these signals.
>
> A: We agree with the reviewer that manually designed signals are not a necessity -- this is exactly the reason that we also proposed an automatic supervision signal search method, trying to find the best supervision signals, i.e., the proxy task. As the reviewer points out, designing a proper proxy task is the root, and our proposed proxy task search has demonstrated its effectiveness and efficiency. We will make this point clear in the revised paper.
> We would also like to provide two reasons that we used “manually designed signals”. (1) In the situations where users don’t want to do or cannot afford proxy-task search, manually designed signals also generalize quite well for different datasets and search spaces, meaning that these signals largely capture the intrinsic characteristics of an ML task, either CV or NLP. (2) Although self-supervised learning works very well, the training and searching still take some effort. ; however, using pre-designed signals can directly guide the search with orders of magnitude of speedup.
>
> Q: I think the correlation comparison in Table 2 is not fair. GenNAS can use combo of signals, but all other methods only measure the correlation on a single dataset. Maybe using combination of CIFAR and ImageNet-16 can further boost their correlation.
>
> A: We would like to clarify a confusion: GenNAS uses combinations of different signal bases, but does not use combinations of different datasets. We only use a batch of images from the target training dataset and the synthetic signal data serves as the groundtruth. All the methods are evaluated on a single dataset, either on CIFAR or ImageNet-16. We believe the comparison in Table 2 is fair.
>
>
>
> Q: Another efficient & w.o. proxy task NAS work is not considered [1].
>
> A: Thank you and we will add the comparison of [1] in the final version. We compare our method with at least 8 efficient NAS methods[2 (ICLR 21), 3 (CVPR 19) ,4 (ICML 21),5 (CVPR 20),6 (ECCV 20),7 (UAI 20), 8 (ICLR 19), 9 (ICCV 19)] which is sufficient to show consistent performance of GenNAS. We provide the NASBench-201 results of our method v.s. [1]  in the following table, and would like to add the results into the later version:
>
> |Methods|CIFAR-10|CIFAR-100|Tiny-ImageNet|Search Cost(hrs)|
> |---|---|---|---|---|
> |TE-NAS|93.9±0.47|71.24±0.56|42.38±0.46|0.43|
> |GenNAS|94.18±0.10|72.56±0.74|45.59±0.54|0.3|
>
> [1] Neural Architecture Search on ImageNet in Four GPU Hours: A Theoretically Inspired Perspective
>
> [2] Abdelfattah, Mohamed S., et al. "Zero-cost proxies for lightweight nas." arXiv preprint arXiv:2101.08134 (2021).
>
> [3] Dong, Xuanyi, and Yi Yang. "Searching for a robust neural architecture in four gpu hours." Proceedings of the IEEE/CVF Conference on Computer Vision and Pattern Recognition. 2019.
>
> [4] Mellor, Joe, et al. "Neural architecture search without training." International Conference on Machine Learning. PMLR, 2021.
>
> [5] Zhou, Dongzhan, et al. "Econas: Finding proxies for economical neural architecture search." Proceedings of the IEEE/CVF Conference on Computer Vision and Pattern Recognition. 2020.
>
> [6] Liu, Chenxi, et al. "Are labels necessary for neural architecture search?." European Conference on Computer Vision. Springer, Cham, 2020.
>
> [7] Li, Liam, and Ameet Talwalkar. "Random search and reproducibility for neural architecture search." Uncertainty in artificial intelligence. PMLR, 2020.
>
> [8] Liu, Hanxiao, Karen Simonyan, and Yiming Yang. "Darts: Differentiable architecture search." arXiv preprint arXiv:1806.09055 (2018).
>
> [9] Dong, Xuanyi, and Yi Yang. "One-shot neural architecture search via self-evaluated template network." Proceedings of the IEEE/CVF International Conference on Computer Vision. 2019.

---

### Official Review · Reviewer_SYQY · 2021-07-12

**Rating:** 3
**Confidence:** 5

**Summary:**

This work made an interesting attempt to solve the generic neural architecture search (NAS) based on regression. A self-supervised regression task based on synthetic signal is proposed to evaluate the performance of a neural network architecture without the true labels of the images as in most previous NAS algorithms, leading to an efficient and effective search scheme. The ranking correlation coefficient, i.e., Spearman coefficient is taken to quantitatively evaluate the efficacy of this novel scheme on extensive search space.

**Ethical Concerns:**




**Ethics Review Area:**

["I don’t know"]

**Limitations And Societal Impact:**

1.  Though for one-shot NAS, the ranking correlation seems to be a good metric for its performance evaluation. There is still some performance gap between the searched one and the stand-alone model. So the authors should better provide more justification on this aspect, that is, why the ranking correlation is enough to validate the effectiveness of their NAS scheme.

2. No strong analysis or no insight on why such a synthetic signal works effectively for evaluation of a network architecture.

3. The authors claimed that the proposed NAS is task agnostic, however, in all the experiments, only classification related experiments are provided to show the performance. As such, this seems not convincing to support such a claim.  More tasks such as detection, segmentation should be provided on this point.

4. Only CIFAR-10 results are reported for end-to-end NAS comparisons, which is not convincing to show the efficacy of the proposed scheme. ImageNet based results should be provided to further support this point.

5. Representation of this manuscript needs a lot of polish.

**Main Review:**

Generally speaking, the whole idea seems a good attempt for the hot topic of NAS, leading to possible new insight to task agnostic NAS. Comprehensive tests across 13 CNN search space and 1 NLP space illustrate a promising ranking correlation between the ground truth and the network evaluated based on regression tasks.  However, some big concerns remain for its current form, which are shown in the below section.

**Time Spent Reviewing:**

about 3 hours

---

> ### Author Response · Authors · 2021-08-10
> **Replies to SYQY**
>
> We thank the reviewer for the very detailed and useful comments. Here we would like to address the reviewer’s questions:
>
> Q: Though for one-shot NAS, the ranking correlation seems to be a good metric for its performance evaluation. There is still some performance gap between the searched one and the stand-alone model. So the authors should better provide more justification on this aspect, that is, why the ranking correlation is enough to validate the effectiveness of their NAS scheme.
>
> A: Thank you for pointing this out. We further add the percentage of the top-10% most-accurate sampled architectures that are predicted correctly to be within the top-10% of the predicted architectures to further support the performance of our method: https://docs.google.com/spreadsheets/d/18OA4qW3-KxsQdUkyJf0Vre4sOBEQKti77sB715AHWa8/edit?usp=sharing .
> For NASBench-101/201/NLP, GenNAS can retrieve more architectures compared to the latest zero-shot and efficient methods [2,4,5,6]. Specifically, for NASBench-101, UNNAS[6] released architectures trained on ImageNet with the same hyperparameter settings. GenNAS is able to retrieve 64% of the top-10 architectures compared to NASWOT (36%), synflow (14%) and 4 different tasks proposed by [6] with 10 epochs’ early stop (39%, 49%, 44%, 46%). For the NDS CIFAR-10, GenNAS achieves comparable results with cls@ep10. It’s important that when the GenNAS task is optimized for the target search space, GenNAS-D retrieves 59% architectures on DARTS search space and GenNAS-R retrieves 64% architectures on ResNet search space, which further support the proxy task search shown in 3.2. Also, for the NDS ImageNet, our method is still robust towards the large dataset. E.g. GenNAS-D retrieves 58% on DARTS imagenet search space and GenNAS-R retrieves 75% and 64% architectures on ResNeXt-A and ResNeXt-B search space respectively. Note that synflow[2] fails on retrieving architectures on ImageNet (<20%).
> Moreover, our end-to-end search results (Table 3) also support the effectiveness of our NAS scheme. Using the ranking correlation score as an evaluation to choose the proxy task, our GenNAS can successfully search for the state-of-the-art stand-alone network architectures.
>
> Q: No strong analysis or no insight on why such a synthetic signal works effectively for evaluation of a network architecture.
>
> A: As we explained in the introduction section, the intuition of using synthetic signals for regression-based proxy tasks is that, these synthetic signals such as 1D/2D sine and dot noise are basic signals that compose a large range of real-world signals. Therefore, learning these basic signals can evaluate the network’s capability of capturing different frequencies, spatial information, and resizing. It is also explained in Section 3.1.2. The synthetic signals serve as the self-supervised task for neural networks to learn. We also conduct the ablation study (Table 1) and it empirically reveals that simple patterns can achieve high ranking correlation. In appendix, Figure 2 a&b suggest that our method achieves better results compared to the classification with the same amount of data with respect to ranking correlation (~0.8 v.s. ~0.4). We will add more explanations and experiments in the revised version that explain why such synthetic signals work effectively.
>
> Q: The authors claimed that the proposed NAS is task agnostic, however, in all the experiments, only classification related experiments are provided to show the performance. As such, this seems not convincing to support such a claim. More tasks such as detection, segmentation should be provided on this point.
>
>
> We appreciate and agree with the reviewer’s comments about enhancing the task-agnostic point, and we would definitely perform more extensive experiments on detection and segmentation in our follow-up experiments. In this paper, we claim it is “task-agnostic” because of three reasons: (1) in both CV and NLP models, the supervising task is a regression task but the target task is classification; (2) the general idea of GenNAS, i.e., using synthetic signals in regression, works well for both CV and NLP tasks; (3) for the classification tasks in CV, the datasets and search spaces are very diverse, but our GenNAS is dataset-agnostic: the searched network architecture on one dataset naturally works well on other datasets. Such features of GenNAS makes it highly adaptive to unseen tasks/datasets.
>
>
> Q: Only CIFAR-10 results are reported for end-to-end NAS comparisons, which is not convincing to show the efficacy of the proposed scheme. ImageNet based results should be provided to further support this point.
>
> A: Thank you for the concern. We report the CIFAR10/CIFAR-100/ImageNet-tiny results for NASBench-201 and PTB results for NASBenchNLP. The main contribution and goal of our work is to show the effectiveness of our regression-based evaluation. Therefore, we deliberately choose to compare with several computationally light NAS works that are friendly for repetition and reproduction. Multiple repeats can be done for each experiment, and we can see the systematic performance gain by different NAS algorithms rather than the engineering gains on ImageNet. Also, for the ImageNet based sample experiments, we report the top-10% most-accurate models that are predicted correctly to be within the top-10% in the above question (across 9 search spaces) to further support the performance of our method. Moreover, we report the searched architectures in https://docs.google.com/spreadsheets/d/14NZu6H5Ua1lEGIgrOkBjcQ_I43l-k1drgQ6VTjEasq8/edit?usp=sharing and all our experiments can be reproduced.
>
> Q: Representation of this manuscript needs a lot of polish.
>
> A: Thank you for the comment. We will further polish the writing for the final version of the paper.
>
> [1] Liu, Chenxi, et al. "Are labels necessary for neural architecture search?." European Conference on Computer Vision. Springer, Cham, 2020.
>
> [2] Duan, Yawen, et al. "TransNAS-Bench-101: Improving Transferability and Generalizability of Cross-Task Neural Architecture Search." Proceedings of the IEEE/CVF Conference on Computer Vision and Pattern Recognition. 2021.
>
> [3] Kornblith, Simon, Jonathon Shlens, and Quoc V. Le. "Do better imagenet models transfer better?." Proceedings of the IEEE/CVF Conference on Computer Vision and Pattern Recognition. 2019.

---

### Official Review · Reviewer_p1m2 · 2021-07-15

**Rating:** 6
**Confidence:** 4

**Summary:**

In this paper, the authors propose a generic NAS method named GenNAS, which uses regression as the self-supervised task. Moreover, the proposed method is task-agnostic and highly efficient. Experiments demonstrate the effectiveness of the proposed method. My detailed comments are as follows.

Positive points:
1.	The authors proposed a new regression-based NAS method.
2.	The results from experiments show the proposed method achieves promising performance.


**Limitations And Societal Impact:**

The limitations and potential negative societal impact of their work should be clarified.

**Main Review:**

Negative points:
1. In section 3.1.1, it is not clear how to divide a network into N stages. More details should be provided.
2. In the last line of Table 3, cls@ep10 method yields better performance than GenNAS-N. More explanations are required. Moreover, it seems that GenNAS should not be highlighted.
3. In section 4.4, line 352, it’s not clear that how to obtain L^*=4.36.
4. Several state-of-the-art NAS methods [1,2] should be compared in the paper.
5. Kendall Tau (KTau) is a widely used metric to measure the ranking ability. It would be stronger to use KTau to compare different methods.
6. Since this paper is a NAS paper, the results of the final searched architectures on CIFAR and ImageNet should be reported.

References:
[1] Proxylessnas: Direct neural architecture search on target task and hardware, ICLR 2019.
[2] Once-for-all: Train one network and specialize it for efficient deployment, ICLR 2020.

--Post Rebuttal--

I am happy to raise my score based on the additional results. I hope that all the results can be included in the paper as promised by the authors.


**Time Spent Reviewing:**

4

---

> ### Author Response · Authors · 2021-08-10
> **Replies to p1m2**
>
> We thank the reviewer for the very detailed and useful comments. Here we would like to address the reviewer’s questions:
>
> Q: In section 3.1.1, it is not clear how to divide a network into N stages. More details should be provided.
>
> A.: Thank you for pointing this out. In the GenNAS, we use 3 stages, and each stage accepts the input images with size 32, 16, and 8, respectively (when using CIFAR-like dataset). We will add a clear explanation in the later version.
>
> Q: In the last line of Table 3, cls@ep10 method yields better performance than GenNAS-N. More explanations are required. Moreover, it seems that GenNAS should not be highlighted.
>
> A: We’ll fix this typo and thanks for pointing it out. We acknowledge that in this one case, cls@ep10 achieves better performance than our results; however, for most of the scenarios, our method achieves better results than cls@ep10.
>
> Q: In section 4.4, line 352, it’s not clear that how to obtain L^*=4.36.
>
> A: The L^* is the best performing architecture (lowest testing log perplexity) as benchmarked in NAS-Bench-NLP in reference [1].  You might refer to the NAS-Bench-NLP paper for more details.
>
> Q: Several state-of-the-art NAS methods should be compared in the paper.
>
> A: Thanks for pointing this out. Since the main focus of our work is to study the effectiveness of our extremely light-weight evaluation and search method, we deliberately choose to study and compare with several existing computationally light works that are friendly for repetition and reproduction. Multiple repeated runs can be done for each experiment, and we can see the systematic performance gain by different NAS algorithms rather than the “computing resource”  gains on ImageNet. We have tried our best to include comparison with 8 most recent efficient SOTA methods [2 (ICLR 21), 3 (CVPR 19) ,4 (ICML 21),5 (CVPR 20),6 (ECCV 20),7 (UAI 20), 8 (ICLR 19), 9 (ICCV 19)].
> Moreover, we show the percentage of the top-10% most-accurate sampled architectures that are predicted correctly to be within the top-10% of the predicted architectures to further support the performance of our method: https://docs.google.com/spreadsheets/d/18OA4qW3-KxsQdUkyJf0Vre4sOBEQKti77sB715AHWa8/edit?usp=sharing
> For NASBench-101/201/NLP/NDS, GenNAS can retrieve more architectures compared to the latest zero-shot and efficient methods [2,4,5,6]. Specifically, for NASBench-101, UNNAS[6] released architectures trained on ImageNet with the same hyperparameter settings. GenNAS is able to retrieve 64% of the top-10 architectures compared to NASWOT (36%), synflow (14%) and 4 different tasks proposed by [6] with 10 epochs’ early stop (39%, 49%, 44%, 46%). For the NDS CIFAR-10, GenNAS achieves comparable results with cls@ep10. It’s important that when the GenNAS task is optimized for the target search space, GenNAS-D retrieves 59% architectures on DARTS search space and GenNAS-R retrieves 64% architectures on ResNet search space, which further support the proxy task search shown in 3.2. Also, for the NDS ImageNet, our method is still robust towards the large dataset. E.g. GenNAS-D retrieves 58% on DARTS imagenet search space and GenNAS-R retrieves 75% and 64% architectures on ResNeXt-A and ResNeXt-B search space respectively. Note that synflow[2] fails on retrieving architectures on ImageNet (<20%). We will add more comparisons into the revised version.
>
> Q: Kendall Tau (KTau) is a widely used metric to measure the ranking ability. It would be stronger to use KTau to compare different methods.
>
> A: The reason we choose the Spearman Rho is that methods like zero-shot NAS[2] didn’t release the code for the NLP-NAS benchmark and only have Spearman Rho results. In order to compare with these existing works, we need to use the same correlation metric (Spearman Roh or Kendall Tau). Nevertheless, we would like to also add KTau ranking for the final version of the paper.
>
> Q. Since this paper is a NAS paper, the results of the final searched architectures on CIFAR and ImageNet should be reported.
>
> A. Thank you for pointing this out. We’ve added all the searched network architectture results in the table https://docs.google.com/spreadsheets/d/14NZu6H5Ua1lEGIgrOkBjcQ_I43l-k1drgQ6VTjEasq8/edit?usp=sharing .
>
> [1] Klyuchnikov, Nikita, et al. "NAS-Bench-NLP: neural architecture search benchmark for natural language processing." arXiv preprint arXiv:2006.07116 (2020).
>
> [2] Abdelfattah, Mohamed S., et al. "Zero-cost proxies for lightweight nas." arXiv preprint arXiv:2101.08134 (2021).
>
> [3] Dong, Xuanyi, and Yi Yang. "Searching for a robust neural architecture in four gpu hours." Proceedings of the IEEE/CVF Conference on Computer Vision and Pattern Recognition. 2019.
>
> [4] Mellor, Joe, et al. "Neural architecture search without training." International Conference on Machine Learning. PMLR, 2021.
>
> [5] Zhou, Dongzhan, et al. "Econas: Finding proxies for economical neural architecture search." Proceedings of the IEEE/CVF Conference on Computer Vision and Pattern Recognition. 2020.
>
> [6] Liu, Chenxi, et al. "Are labels necessary for neural architecture search?." European Conference on Computer Vision. Springer, Cham, 2020.
>
> [7] Li, Liam, and Ameet Talwalkar. "Random search and reproducibility for neural architecture search." Uncertainty in artificial intelligence. PMLR, 2020.
>
> [8] Liu, Hanxiao, Karen Simonyan, and Yiming Yang. "Darts: Differentiable architecture search." arXiv preprint arXiv:1806.09055 (2018).
>
> [9] Dong, Xuanyi, and Yi Yang. "One-shot neural architecture search via self-evaluated template network." Proceedings of the IEEE/CVF International Conference on Computer Vision. 2019.

---

> > ### Comment · Reviewer_p1m2 · 2021-08-31
> > **Post Rebuttal**
> >
> > Thanks for the responses from the authors! After reading the rebuttal, I still have several concerns.
> >
> > 1. The very important results measured by Kendall Tau are still missing. I do not think that being unable to compare with [2] is an essential reason to not use Kendall Tau. In my opinion, most NAS papers use Kendall Tau for comparisons.
> >
> > 2. There seem too many experimental results which "will be included or reported".
> >
> > Do the authors obtain some of these results currently? The possible new results may help me finalizing my rating.

---

> > > ### Author Response · Authors · 2021-09-01
> > > **Further responses**
> > >
> > > Thank you so much again for your reply and willingness to reevaluate our work.
> > >
> > > ### Q1: Kendall Tau v.s. Spearman
> > >
> > > We do not think that most previous works adopted Kendall Tau. We found that [1,2,3,4,5,6,7,8] use Spearman Rho and works [9,10,11] use Kendall Tau. That have been said, we provide the full comparison based on Kendall Tau in the following link:
> > >
> > > https://docs.google.com/spreadsheets/d/1KnSo3pTQLtM5GbSLD4FaH9toZrvAFJAEfLYra_hv8AE/edit?usp=sharing
> > >
> > > ### Q2: There seem too many experimental results which "will be included or reported".
> > >
> > > We can put all these additional results we have got into the appendix. We see no wrong to do this as it can benefit readers like you that are interested in all these details.
> > >
> > > ### Q3: Additional results?
> > >
> > > Since we fortunately have access to more computational resources during the rebuttal session, we are able to evaluate the searched architecture on ImageNet and report the results. Our reported architecture is searched on the DARTS search space with 16 random images from CIFAR-10, where the search time is just 1 GPU hour on a single 1080 TI GPU. Our searched architecture is on par with the architectures searched on ImageNet (e.g. UNNAS[2], PC-DARTS[13], ProxylessNAS[12]).
> > >
> > >
> > > | Architecture            | Top-1 Err. | Top-5 Err. | Params(M) | FLOPS | Search cost  (GPU days) | Search Method |
> > > |-------------------------|------------|------------|-----------|-------|-------------------------|---------------|
> > > | NASNet-A                | 26.0       | 8.4        | 5.3       | -     | 2000                    | RL            |
> > > | AmoebaNet-C             | 24.3       | 7.6        | 6.4       | -     | 3150                    | EA            |
> > > | DARTS(2nd)              | 26.7       | 8.7        | 4.7       | 574   | 4.0                     | gradient      |
> > > | P-DARTS                 | 24.4       | 7.4        | 4.9       | 557   | 0.3                     | gradient      |
> > > | PC-DARTS (CIFAR10)      | 25.1       | 7.8        | 5.3       | 586   | 0.1                     | gradient      |
> > > | PC-DARTS (ImageNet)       | 24.2       | 7.3        | 5.3       | 597   | 3.8                     | gradient      |
> > > | ProxylessNAS (ImageNet)        | 24.9       | 7.5        | 7.1       | -     | 8.3                     | gradient      |
> > > | UNNAS  (ImageNet) | 24.1       | -          | 5.1       | 559  | 2                       | gradient      |
> > > | UNNAS (Reproduced)      | 24.5       | 7.4        | 5.1      | 559   | 2                       | gradient      |
> > > | GenNAS                  | 24.4       | 7.5        | 4.9       | 554   |0.04* | EA + few-shot |
> > >
> > > *For GenNAS, 0.2 GPU days on a single 1080i needs to be added if we include the time of proxy task search, though as we have mentioned, the regression-based GenNAS-COMBO itself without proxy task search is already significant (the experiment is running).
> > >
> > > [1] Ying, Chris, et al. "Nas-bench-101: Towards reproducible neural architecture search." International Conference on Machine Learning. PMLR, 2019.
> > >
> > > [2] Liu, Chenxi, et al. "Are labels necessary for neural architecture search?." European Conference on Computer Vision. Springer, Cham, 2020.
> > >
> > > [3] Zela, Arber, Julien Siems, and Frank Hutter. "Nas-bench-1shot1: Benchmarking and dissecting one-shot neural architecture search." arXiv preprint arXiv:2001.10422 (2020).
> > >
> > > [4] Mehrotra, Abhinav, et al. "NAS-Bench-ASR: Reproducible Neural Architecture Search for Speech Recognition." International Conference on Learning Representations. 2020.
> > >
> > > [5] Abdelfattah, Mohamed S., et al. "Zero-cost proxies for lightweight nas." arXiv preprint arXiv:2101.08134 (2021).
> > >
> > > [6] Li, Changlin, et al. "Bossnas: Exploring hybrid cnn-transformers with block-wisely self-supervised neural architecture search." arXiv preprint arXiv:2103.12424 (2021).
> > >
> > > [7] Dubatovka, Alina, et al. "Ranking architectures using meta-learning." arXiv preprint arXiv:1911.11481 (2019).
> > >
> > > [8] Ru, Binxin, et al. "Interpretable Neural Architecture Search via Bayesian Optimisation with Weisfeiler-Lehman Kernels." arXiv preprint arXiv:2006.07556 (2020).
> > >
> > > [9] Mellor, Joe, et al. "Neural architecture search without training." International Conference on Machine Learning. PMLR, 2021.
> > >
> > > [10] Zhou, Dongzhan, et al. "Econas: Finding proxies for economical neural architecture search." Proceedings of the IEEE/CVF Conference on Computer Vision and Pattern Recognition. 2020.
> > >
> > > [11] Zhang, Miao, et al. "One-shot neural architecture search: Maximising diversity to overcome catastrophic forgetting." IEEE Transactions on Pattern Analysis and Machine Intelligence (2020).
> > >
> > > [12] Cai, Han, Ligeng Zhu, and Song Han. "Proxylessnas: Direct neural architecture search on target task and hardware." arXiv preprint arXiv:1812.00332 (2018).
> > >
> > > [13] Xu, Yuhui, et al. "PC-DARTS: Partial channel connections for memory-efficient architecture search." arXiv preprint arXiv:1907.05737 (2019).

---

> > > > ### Comment · Reviewer_p1m2 · 2021-09-02
> > > > **Re: Further responses**
> > > >
> > > > Thanks for the further responses. The authors indeed provide more results to verify their method. Most of my concerns have been addressed. I would like to raise my rating.

---

> > > > > ### Author Response · Authors · 2021-09-02
> > > > > **Thank you for your efforts.**
> > > > >
> > > > > We would like to thank the reviewers for their thoughtful comments and efforts towards improving our manuscript.

---

### Official Review · Reviewer_aBAs · 2021-07-16

**Rating:** 6
**Confidence:** 4

**Summary:**

This article introduces a novel low-cost regression proxy task, which has a high correlation with the test performance of the final model. Its contributions include the following:

1. Introducing a novel low-cost regression agent task, which greatly reduces the cost of evaluation.
2. This proxy task is basically independent of downstream tasks, and can make the searched model achieve approximately consistent results in multiple search spaces and datasets.
3. Competitive results can be achieved in End-to-End NAS.

**Main Review:**

+ **Originality**: This article designs a very interesting proxy task, which allows the network to fit a set of hand-designed bases through regression to determine whether the network performance is good or bad.
+ **Quality**:  There are a few questions:
1. In the proxy task, why the input of CNN is unlabeled real data, but the input of RNN is random data?
2. The image does contain information of different frequencies, so the basis of the design looks more reasonable for the image. But for RNN, is it reasonable to choose the same base?

+ **Clarity**: This article is basically written in a well-organized manner. From the introduction of proxy task, to why such a proxy task was designed, to experimental verification and proxy task improvement (proxy task search), the whole logical line is very clear.

+ **Significance**: Designing a low-cost proxy task is very important for NAS. The proxy task designed in this article uses a set of bases that have nothing to do with downstream data to evaluate the performance of the network at a very low cost. This is relatively novel. Previous work focused on network structure and data. This work introduces a new direction.

**Time Spent Reviewing:**

4 hours

---

> ### Author Response · Authors · 2021-08-10
> **Replies to aBAs**
>
> We thank the reviewer for the very detailed and useful comments. Here we would like to address the reviewer’s questions:
>
> Q: In the proxy task, why the input of CNN is unlabeled real data, but the input of RNN is random data?
>
> A:
> We drew an illustration of the RNN regression which can be found here: https://drive.google.com/file/d/1j4IzfRyBpLrfSLmjr12QzS5e-RYu-e2l/view?usp=sharing
> The left figure shows the RNN model for NLP many-to-one tasks on the PTB dataset. The words are projected on an embedding layer first, and the prediction comes from a fully connected layer with the output size as the dictionary length. In GenNAS, the embedding layer is replaced with the fully connected layer, and both the inputs and the groundtruth of RNN are synthetic tensors generated by the configuration with shape [l,b,d] where l is the length of the sequence and b is the batch size and d is the input dimension. We are sorry about the misunderstanding and we will change the “random tensor” to “synthetic tensor” in page 4 #163.
> The reason to use synthetic tensors for RNN inputs is that the language data is hard to integrate into the input since we removed the embedding layer. Moreover, we want to emphasize that, we are the first to reveal that learning the synthetic signal instead of using real data in a few-shot fashion can evaluate the RNN efficiently with high correlation (0.81) (see Table 2 & 3), which is also a contribution.
>
> Q: The image does contain information of different frequencies, so the basis of the design looks more reasonable for the image. But for RNN, is it reasonable to choose the same base?
>
> A: For RNNs, we consider: 1. The signals along the d dimension (input dimension & output dimension) are 1D signals so it contains information of 1D frequencies. 2. The signals along the l dimension (sequence length) require time-series prediction for RNNs to have stable internal memory, which is also 1D signals that are orthogonal with d dimension. Therefore, RNN signals are also 2D signals so we believe it is reasonable to elaborate the 2D image along the d and l (as shown in the illustration above) dimensions. We will add more explanations in the revised version.

---

> > ### Comment · Reviewer_aBAs · 2021-08-29
> > **Post Rebuttal**
> >
> > Thanks for your reply.
> >
> > I still have some concerns about this work.
> >
> > 1. The reasons for not using raw data as input seem to make sense. But this raises another concern: why not also use synthetic data as input for CNN? It looks more reasonable that CNN & RNN take a unified-form data as input, doesn't it?
> > 2. Regarding the question of why RNN uses the same synthetic signal as CNN, I don’t think the explanation you gave is convincing. Why the 1D signals must contain information of 1D frequencies?
> > 3. Taking into account the concern of other reviewers about Spearman $\rho$ , it is a pity that you didn’t provide relevant data to further prove its effectiveness. It won't take a lot of time. I think you should provide them.
> >
> > I hope you can give a convincing and reasonable response.

---

> > > ### Author Response · Authors · 2021-08-29
> > > **To Post Rebuttal**
> > >
> > > Q: The reasons for not using raw data as input seem to make sense. But this raises another concern: why not also use synthetic data as input for CNN? It looks more reasonable that CNN & RNN take a unified-form data as input, doesn't it?
> > >
> > > A: Actually, our method is rather stable no matter whether the input is a synthetic signal or true signal. The output using the signal bases is more crucial. By using the synthetic data as the input for CNN, we still can achieve 0.79 Spearman Rho on NASBench-101. We will include the experiments in the updated version.
> > >
> > >
> > > Q: Regarding the question of why RNN uses the same synthetic signal as CNN, I don’t think the explanation you gave is convincing. Why the 1D signals must contain information of 1D frequencies?
> > >
> > > A: The reviewer seems to be unfamiliar with the functional fourier representation. Actually, any functions in compact L2 space can be well approximated by a linear combination of fourier bases. Or in another word, linear combinations of fourier bases are dense in any L2-continuous functional space due to Stone-Weierstrass theorem applied to fourier bases. We refer some advanced calculus or functional analysis tutorials to the reviewer.
> > >
> > >  This is also the key that makes our regression-based NAS work so generally. The neural architecture does not need to classify the examples in a particular training datasets. The neural architecture just needs to fit the bases that are expressive enough to represent any functions.
> > >
> > > Q: Taking into account the concern of other reviewers about Spearman Rho, it is a pity that you didn’t provide relevant data to further prove its effectiveness. It won't take a lot of time. I think you should provide them.
> > >
> > > A:
> > >
> > > The reviewer seems unfamiliar with the relation between Spearman Rho and Kendall Tau. There is a nice sandwich equality to make sure that these two metrics are compatible [9,10,11] . It really does not matter to choose to use which metric as long as all models use the same metric. We do not think Reviewer p1m2 gives a fair argument by asking for this. Moreover, previous methods keep using Spearman Rho [1,2,3,4,5,6,7,8]. It is impossible for us to rerun all the baseline models to collect the results using new metrics within such a short time.
> > >
> > > We have provided the detailed experimental results in both the Appendix and through links in the responses to the review comments. We’ve already made the paper self-contained.
> > >
> > > [1] Ying, Chris, et al. "Nas-bench-101: Towards reproducible neural architecture search." International Conference on Machine Learning. PMLR, 2019.
> > >
> > >
> > > [2] Liu, Chenxi, et al. "Are labels necessary for neural architecture search?." European Conference on Computer Vision. Springer, Cham, 2020.
> > >
> > > [3] Zela, Arber, Julien Siems, and Frank Hutter. "Nas-bench-1shot1: Benchmarking and dissecting one-shot neural architecture search." arXiv preprint arXiv:2001.10422 (2020).
> > >
> > > [4] Mehrotra, Abhinav, et al. "NAS-Bench-ASR: Reproducible Neural Architecture Search for Speech Recognition." International Conference on Learning Representations. 2020.
> > >
> > > [5] Abdelfattah, Mohamed S., et al. "Zero-cost proxies for lightweight nas." arXiv preprint arXiv:2101.08134 (2021).
> > >
> > > [6] Li, Changlin, et al. "Bossnas: Exploring hybrid cnn-transformers with block-wisely self-supervised neural architecture search." arXiv preprint arXiv:2103.12424 (2021).
> > >
> > > [7] Dubatovka, Alina, et al. "Ranking architectures using meta-learning." arXiv preprint arXiv:1911.11481 (2019).
> > >
> > > [8] Ru, Binxin, et al. "Interpretable Neural Architecture Search via Bayesian Optimisation with Weisfeiler-Lehman Kernels." arXiv preprint arXiv:2006.07556 (2020).
> > >
> > > [9] Colwell, D. J., and J. R. Gillett. “66.49 Spearman versus Kendall.” The Mathematical Gazette, vol. 66, no. 438, 1982, pp. 307–309. JSTOR, www.jstor.org/stable/3615525. Accessed 29 Aug. 2021.
> > >
> > > [10] Fredricks, Gregory A., and Roger B. Nelsen. "On the relationship between Spearman's rho and Kendall's tau for pairs of continuous random variables." Journal of statistical planning and inference 137.7 (2007): 2143-2150.
> > >
> > > [11] Trutschnig, Wolfgang, and Thomas Mroz. "A sharp inequality for Kendall’s τ and Spearman’s ρ of Extreme-Value Copulas." Dependence Modeling 6.1 (2018): 369-376.

---

> > > > ### Comment · Reviewer_aBAs · 2021-08-30
> > > > **Re: To Post Rebuttal**
> > > >
> > > > Thank you for your **kind** reply!
> > > >
> > > > 1. What I concern with is why this method doesn’t take synthetic data as the input of the network at the beginning, which seems more reasonable, doesn't it? In addition, if the input of the model is synthetic data, how to design the corresponding regression target? The same as the input?
> > > > 2. Your reply to me this time is quite convincing.
> > > > 3. Spearman’s rho and Kendall’s tau are two commonly used indicators to describe correlation. Sandwich inequality does not mean that the two are positively correlated. In fact, the two describe different aspects of the data. But it somewhat makes sense for all models to use the same metric.
> > > >
> > > > Please give a reasonable explanation  towards my first concern **friendly**.

---

> > > > > ### Comment · Area_Chair_gZ6e · 2021-08-30
> > > > > **Spearman vs. Kendall Tau**
> > > > >
> > > > > I don't understand this ask for both Spearman's vs. Kendall Tau. If I know Spearman's correlation, using the inequality I can get the Kendall Tau's lower bound or vice versa (I think this is what the authors mean, right?)
> > > > >
> > > > > What is the different aspect that is captured by Kendall Tau that Spearman's does not capture that is relevant here? (Or vice versa?)

---

> > > > > > ### Comment · Reviewer_aBAs · 2021-08-31
> > > > > > **Re: Spearman vs.Kendall Tau**
> > > > > >
> > > > > > Hi!
> > > > > >
> > > > > > Indeed, both are indicators of correlation. However, according to the calculation (one can refer to the appendix of [GreedyNas](https://arxiv.org/abs/2003.11236) for more details), the former is more sensitive to discrepancies in data while the latter focuses on concordant and discordant pairs, not the absolute difference, thus is less sensitive.
> > > > > >
> > > > > > Besides, I think what the author want to express is the two are equivalent or positive correlation, which can't be derived according to the sandwich inequality. Here is an example (without taking into account the pValue):
> > > > > > ```
> > > > > > a = [0, 1, 2, 3, 4, 5, 6]
> > > > > > b = [0, 2, 6, 1, 3, 4, 5]
> > > > > > c = [2, 1, 0, 5, 3, 4, 6]
> > > > > > rho(a, b) = 0.57, tau(a, b) = 0.52
> > > > > > rho(a, b) = 0.75, tau(a, b) = 0.52
> > > > > > ```
> > > > > > According to the case, b and c take the same tau but they take different rho with a clear difference.
> > > > > >
> > > > > > The reason why I hope the author provides tau data is to further confirm the effectiveness of the method. But as the author said, some papers indeed only take Spearman rho as the metric. Therefore It's OK that the author doesn't provide these data.

---

> > > > > > > ### Comment · Area_Chair_gZ6e · 2021-08-31
> > > > > > > **Nice!**
> > > > > > >
> > > > > > > Thanks for that insightful response. Nice toy example as well!

---

> > > > > ### Author Response · Authors · 2021-09-01
> > > > > **Response to the question 1: Synthetic data as the input and comparing metrics**
> > > > >
> > > > > Thank you so much for your interest and further in-depth questions!
> > > > >
> > > > > Actually, we have tried both using synthetic data and real images as the input of the network. As using the later one achieves slightly better performance, we just kept the later one and did not do further change. When we used the synthetic data as the input, we did not directly use the input synthetic data as the output for regression. Instead, we did random combinations of synthetic data as the input signal and the output supervision. Thank you for your suggestions! we consider also reporting the experiments that use the synthetic data as input to illustrate the full picture.
> > > > >
> > > > > Regarding the evaluation metrics Spearman's v.s. Kendall tau's:
> > > > >
> > > > > Thank you for constructing a very nice example. The AC’s understanding of equivalent metrics is correct. When one talks about the equivalence between two norms/metrics, it just means that two norms/metrics follow a sandwich bound: For example, in a finite dimensional space, a well known result is that the l1 norm is equivalent to the l2 norm, of which both can be used to characterize the properties of algorithms, e.g., convergence rate, though these two metrics are not exactly the same. We can see many pairs of metrics that are equivalent but not exactly the same. An important example is L1 v.s. L2, e.g. in 2-d space:
> > > > >
> > > > > We consider three points:
> > > > > A: (0,0) B: (0,1) C: (0.5,0.5)
> > > > >
> > > > > The L1 distance follows:
> > > > > L1: A to B is 1; A to C is 1
> > > > >
> > > > > The L2 distance follows
> > > > > L2: A to B is 1; A to C is sqrt(2)/2
> > > > >
> > > > > That has been said, we tries our best to borrow the machine and compute the Kendall Tau results, which are showed in the following link. Still, our methods are the best.
> > > > >
> > > > > https://docs.google.com/spreadsheets/d/1KnSo3pTQLtM5GbSLD4FaH9toZrvAFJAEfLYra_hv8AE/edit?usp=sharing

---

> > > > > > ### Comment · Reviewer_aBAs · 2021-09-02
> > > > > > **Thank you for your reply these days！**
> > > > > >
> > > > > > Thank you for your replies these days and the more convincing experimental results！
> > > > > >
> > > > > > The only thing I am currently worried about is the input and output of CNN. If the input and output are separate random synthetic data, I think it still lacks a certain degree of interpretability. It seems more reasonable if the two are the same (maybe I don't understand it enough). I hope you can give a more detailed explanation in the final version, which will make this paper more perfect.
> > > > > >
> > > > > > Based on your recent reply, I am willing to change back to my rating.

---

> > > > > > > ### Author Response · Authors · 2021-09-02
> > > > > > > **Thank you for your comments and suggestions**
> > > > > > >
> > > > > > > Thanks for taking your time to give comments and suggestions. We will also test the CNN performance when the input and output are set the same. We did not do that because we had a concern that it would reduce to nothing but an identity mapping, which is hard to test the architectures.

---

### Comment · Area_Chair_gZ6e · 2021-08-30
**Clarification questions**

Dear Authors,

I just have some clarification questions:

1. In Table 2, on all the different NAS search spaces, how did you obtain final groundtruth test accuracies in order to compute Spearman's? Since these are not part of the set of NAS benchmarks one cannot simply look those values up. Did you train them from scratch? How many architectures from each search space were used? (Sorry if these details are in there somewhere, this is quite an experiment-rich paper :-))

2. In Table 2, GenNAS-N gpu hours reported: do you include the time to conduct proxy task search on this? This is quite important since additional time is hidden away otherwise.

3. Why no gpu hours reported for NDS?

---

> ### Author Response · Authors · 2021-09-01
> **Response**
>
> ### Q1: Where can we get the ground-truth testing accuracy for Table 2?
>
> We assume that the area chair was asking for the testing accuracies of the architectures used in in the sub-table Neural Design Space of Table 2. Because the testing accuracy of other search spaces were provided by benchmarks. Actually, the ground-truth accuracy of  the sub-table Neural Design Space is provided by NDS [1]. Please refer to https://github.com/facebookresearch/nds or more specific downloadable link: https://dl.fbaipublicfiles.com/nds/data.zip that contains such information.
> The table below shows different search spaces and the numbers of sampled architectures in those corresponding search spaces that are evaluated by NDS[1].
>
> | CIFAR-10 | DARTS | DARTS-f | Amoeba | ENAS | NASNet | PNAS | PNAS-f    | ResNet    | ResNeXt-A | ResNeXt-B |
> |----------|-------|---------|--------|------|--------|------|-----------|-----------|-----------|-----------|
> | Number   | 5000  | 5000    | 4983   | 4999 | 4846   | 4999 | 4559      | 25000     | 24999     | 25508     |
> | ImageNet | DARTS | DARTS-f | Amoeba | ENAS | NASNet | PNAS | ResNeXt-A | ResNeXt-B | NB101     |           |
> | Number   | 121   | 499     | 124    | 117  | 122    | 119  | 130       | 164       | 500       |           |
>
>
> ### Q2: In Table 2, GenNAS-N gpu hours reported: do you include the time to conduct proxy task search on this?
>
> First, we want to emphasize that our regression-based approach (GenNAS-COMBO) even without proxy task search has already achieved extremely good performance as shown in the above top-K bins results.
>
> Indeed table 2 does not include the time to conduct proxy task search. We forgot to do so because we viewed the proxy task search as a preprocessing procedure which can be done on a small subset of a search space and datasets, and transferred to other much larger search spaces and datasets. Therefore, we did not count the time cost there and would definitely clarify this in the revised version. Here we provide the time cost of performing proxy task search done on a single GPU 1080i: For the NASBench-101 (GenNAS-N), the search time is 5.75 GPU hours; for the ResNeXt (GenNAS-R), the search time is 4 GPU hours; for DARTS(GenNAS-D), the search time is 12.25 GPU hours.
>
> Note that even after including the above preprocessing time, we may make the following comparison with baselines: For example, we compare with ECONAS[4] which trains each architecture for 15 epoch, if we include the proxy task search time and compare 1000 architectures, the speedup is 6X. If we compare 10000 architectures, the speedup is 32X.
>
> ### Q3: Why no gpu hours reported for NDS?
>
> We assume that the area chair was asking for GPU hours used to train the sampled architectures in the NDS benchmark. Actually we directly use benchmarks provided by the previous NDS github repository (also attached here https://github.com/facebookresearch/nds).
>
> The reason why we do not include such GPU hours is the same as the answer to the previous question. The proxy task search is a preprocessing procedure that may be used or not. The regression method without proxy task search (‘GenNAS-COMBO’) is already very strong (e.g. achieves 60% Top-5% overlap and 0.85 Spearman Rho on NASBench-101), which is the key contribution of this work. We would definitely clarify this point in the revised version.
>
> Even if we adopt proxy task search to better the performance, it can be done on a small subset of a search space and datasets, and later transferred to other much larger search spaces and datasets. In our work, the proxy task search just trains 20 architectures from a certain search space on a small dataset like CIFAR-10.
>
> We have demonstrated the search-space-wise transferability: For example, GenNAS-D task (please refer to Table 2) which is searched on the DARTS search space can achieve state-of-the-art performance on other cell-based search spaces (Amoeba[7], ENAS[8] etc.)
>
> We have demonstrated the dataset-wise transferability: We have even recently evaluated the searched architecture on the huge ImageNet dataset (listed in the following table): GenNAS-D can have robust prediction performance with architectures on ImageNet. Our CIFAR-10, DARTS-search-space searched architecture achieves on-the-par results compared with those works that directly search on ImageNet (e.g. PC-DARTS[6], ProxylessNAS[5], UNNAS[2]). Note that even if we include the GPU time to perform proxy task search + architecture training to get test accuracy used in the search, we are still at least 5-10X  faster than these strong baselines for ImageNet.
>
>
> | Architecture            | Top-1 Err. | Top-5 Err. | Params(M) | FLOPS |
> |-------------------------|------------|------------|-----------|-------|
> | PC-DARTS (ImageNet)       | 24.2       | 7.3        | 5.3       | 597   |
> | ProxylessNAS (ImageNet)  | 24.9       | 7.5        | 7.1       | -     |
> | UNNAS  (ImageNet) | 24.1       | -          | 5.1       | 559   |
> | UNNAS (Reproduced)      | 24.5       | 7.4        | 5.1       | 559    |
> | GenNAS                  | 24.4       | 7.5        | 4.9       | 554   |
>
>
>
> [1] Radosavovic, Ilija, et al. "On network design spaces for visual recognition." Proceedings of the IEEE/CVF International Conference on Computer Vision. 2019.
>
> [2] Liu, Chenxi, et al. "Are labels necessary for neural architecture search?." European Conference on Computer Vision. Springer, Cham, 2020.
>
> [3] https://github.com/facebookresearch/pycls
>
> [4] Zhou, Dongzhan, et al. "Econas: Finding proxies for economical neural architecture search." Proceedings of the IEEE/CVF Conference on Computer Vision and Pattern Recognition. 2020.
>
> [5] Cai, Han, Ligeng Zhu, and Song Han. "Proxylessnas: Direct neural architecture search on target task and hardware." arXiv preprint arXiv:1812.00332 (2018).
>
> [6] Xu, Yuhui, et al. "PC-DARTS: Partial channel connections for memory-efficient architecture search." arXiv preprint arXiv:1907.05737 (2019).
>
> [7] Real, Esteban, et al. "Regularized evolution for image classifier architecture search." Proceedings of the aaai conference on artificial intelligence. Vol. 33. No. 01. 2019.
>
> [8] Pham, Hieu, et al. "Efficient neural architecture search via parameters sharing." International Conference on Machine Learning. PMLR, 2018.

---

> > ### Comment · Area_Chair_gZ6e · 2021-09-01
> > **Thanks and noted!**
> >
> > Thanks again for the prompt detailed response!
> >
> > The NDS dataset of 100k architectures is definitely very nicely utilized by this work! Now I understand how test accuracies on all these search spaces were obtained! (Of course one can train them fully themselves but it is a lot of compute!)
> >
> > Please do include the added proxy search add time to the manuscript. This is an important point. One could even have a breakdown of the time (X hours for search and Y hours for the proxy task search) in the footnote to make it clear why the time maybe higher than expected.
> >
> > I think moving forward including all these details that came out in discussion will be quite useful to include in the manuscript. Some details like KD may be reported either as an alternate column wherever possible or in appendix so as not to distract from the main storyline.

---

> > > ### Author Response · Authors · 2021-09-01
> > > **Many thanks for your appreciation**
> > >
> > > Thank you so much for your prompt check and response.
> > >
> > > We will definitely include the proxy search time into the revised version so that it is more clear for practitioners to use our work.
> > >
> > > We will also include new experiments and discussions in the revised version.
> > >
> > > Just let you know. We are now working hard to evaluate the searched model (GenNAS-COMBO, the one without proxy task search) on ImageNet. We will post the results shortly.

---

### Comment · Area_Chair_gZ6e · 2021-08-30
**A small ask**

Can the authors provide for Nasbench 101 a precision@k breakdown over top-K bins of architectures? See https://arxiv.org/pdf/2008.03064.pdf for exact definition or https://arxiv.org/abs/2106.04010 who term it 'common ratio'.

No need to do it for everything just over nasbench101.

---

> ### Author Response · Authors · 2021-09-01
> **Results on Top-K bins**
>
> Thank the area chair so much for the great interest in our work and results. The overlapping results of top-K bins are listed in the table below. First, let us clarify our understanding of what AC asked for. Please let us know if there is some misunderstanding: We list the ratio between {# of the architectures that are ranked in the top-K bin by both our method and their ground truth testing performance} and {# of architectures that are ranked in the top-K bin according to only their ground truth testing performance}
>
> Following the FEAR[1], we randomly sampled 1000 architectures from the NASBench-101 search space.
>
> GenNAS-COMBO is the one that leverages only regression supervision without proxy task search (please refer to Section 3.1.2) and GenNAS-NB101 is the proxy task searched on NB101 with 20 sampled architectures:
>
>
> | Method       | Top-5% | Top-10% | Top-20% | Top-30% | Top-40% | Top-50% |
> |--------------|--------|---------|---------|---------|---------|---------|
> | GenNAS-COMBO | 0.6    | 0.6     | 0.73    | 0.74    | 0.79    | 0.82    |
> | GenNAS-NB101 | 0.56   | 0.58    | 0.66    | 0.76    | 0.80    | 0.85    |
>
>
> The above results are actually very interesting because NAS based on simple regression yields even slightly better precision than NAS based proxy-task search for small K’s. This suggests that our regression idea is rather crucial. We will definitely include this result in the updated manuscript. Note that the above results cannot be directly compared with the results in Fig. 1 of [2]: [2] only performs NAS among a subset of NASBench-101 because it adopts a differentiable search mechanism which cannot be applied to the whole NAS-Bench-101.
>
> Also we want to draw AC’s attention to our previous response to Reviewer p1m2, where we reported Top-10% for more search spaces and also the performance of baselines in the link below. Green scores are the pure regression ones (without proxy task search) that outperform baselines. Red scores are the best performance. The NASBench-101 Top-10%  is different from the above table. Because we use the same 500-sample architecture set as the UNNAS[3] used in the experiments of the following link.
>
> As it shows, again, our proposed methods significantly outperform all baselines. Even simple regression has brought us the crucial benefit, though proxy-task-search may achieve even better performance.
>
> https://docs.google.com/spreadsheets/d/18OA4qW3-KxsQdUkyJf0Vre4sOBEQKti77sB715AHWa8/edit?usp=sharing
>
>
>
> [1] Dey, Debadeepta, Shital Shah, and Sebastien Bubeck. "FEAR: A Simple Lightweight Method to Rank Architectures." arXiv preprint arXiv:2106.04010 (2021).
>
> [2], Ning, Xuefei, et al. "Evaluating Efficient Performance Estimators of Neural Architectures
> ." arXiv preprint arXiv:2008.03064 (2021).
>
> [3] Liu, Chenxi, et al. "Are labels necessary for neural architecture search?." European Conference on Computer Vision. Springer, Cham, 2020.

---

> > ### Comment · Area_Chair_gZ6e · 2021-09-01
> > **Thanks and noted!**
> >
> > Thanks for your quick response and doing the top-k bin report. It is indeed good that just the pure regression without the task-specific search works well.
> >
> > If authors get time please see my other set of clarification questions.

---

> > > ### Author Response · Authors · 2021-09-01
> > > **Many thanks for your appreciation**
> > >
> > > Thank you so much for your prompt check and response.

---

### Decision · Program_Chairs · 2021-09-27

**Decision:**

Accept (Spotlight)

**Comment:**

This paper proposes a novel approach to ranking architectures using only one minibatch of data by constructing unsupervised regression tasks for CNNs and RNNs via passing the inputs through manually designed feature extractors. The authors have shown the efficacy of the results on a large number of search spaces and baselines.

During discussion period a number of extra results and nuances have been elicited. The authors are strongly recommended to include these in the manuscript. Examples include the precision@K bin of architectures, Kendall-Tau scores and also properly reporting the GenNAS-N gpu hours since it otherwise hides the time for the extra task-specific search.

This ranking method can be quite valuable to the community, so it is strongy recommended that the authors  release high-quality code such that subsequent papers on the topic can utilize this very well.